# EGCN: Entropy-based graph convolutional network for anomalous pattern detection and forecasting in real estate markets

**Dat Le** [1], **Sutharshan Rajasegarar** [1], **Wei Luo** [1], **Thanh Thi Nguyen** [2], **Nhi Vo** [3], **Quang Nguyen** [4], **Maia Angelova** [5] *

1 School of Information Technology, Deakin University, Geelong, Victoria, Australia, 2 Faculty of Information Technology, Monash University, Clayton, Victoria, Australia, 3 Other Bets Division, Automattic Inc., San Francisco, California, United States of America, 4 Investment Department, Mitsubishi Estate Co., Ltd., Chiyoda-ku, Tokyo, Japan, 5 Sir Peter Rigby Digital Futures Institute, Aston University, Birmingham, United Kingdom

☺ These authors contributed equally to this work.
* m.angelova@aston.ac.uk

**Data availability statement:** The data underlying the results presented in the study

## Abstract

Real estate markets are inherently dynamic, influenced by economic fluctuations, policy changes and socio-demographic shifts, often leading to emergence of anomalous—regions, where market behavior significantly deviates from expected trends. Traditional forecasting models struggle to handle such anomalies, resulting in higher errors and reduced prediction stability. In order to address this challenge, we propose EGCN, a novel cluster-specific forecasting framework that first detects and clusters anomalous regions separately from normal regions, and then applies forecasting models. This structured approach enables predictive models to treat normal and anomalous regions independently, leading to enhanced market insights and improved forecasting accuracy. Our evaluations on the UK, USA, and Australian real estate market datasets demonstrates that the EGCN achieves the lowest error both anomaly-free (baseline) methods and alternative anomaly detection methods, across all forecasting horizons (12, 24, and 48 months). In terms of anomalous region detection, our EGCN identifies 182 anomalous regions in Australia, 117 in the UK and 34 in the US, significantly more than the other competing methods, indicating superior sensitivity to market deviations. By clustering anomalies separately, forecasting errors are reduced across all tested forecasting models. For instance, when applying Neural Hierarchical Interpolation for Time Series Forecasting, the EGCN improves accuracy across forecasting horizons. In short-term forecasts (12 months), it reduces MSE from 1.3 to 1.0 in the US, 9.7 to 6.4 in the UK and 2.0 to 1.7 in Australia. For mid-term forecasts (24 months), EGCN achieves the lowest errors, lowering MSE from 3.1 to 2.3 (US), 14.2 to 9.0 (UK), and 4.5 to 4.0 (Australia). Even in long-term forecasts (48 months), where error accumulation is common, EGCN

were obtained from publicly available sources. Australian real estate data were obtained from Australian Property Monitors via AURIN (https://data.aurin.org.au/). Real estate data for the United States were sourced from Zillow (https://www.zillow.com/), and UK property transaction data were sourced from HM Land Registry (https://www.gov.uk/government/organisations/land-registry). Sentiment analysis data were derived from publicly available news headlines using a pre-trained RoBERTa model. All relevant processed data supporting the findings are within the manuscript. https://figshare.com/articles/dataset/EGCN_Entropy-based_Graph_Convolutional_Network_for_Anomalous_Pattern_Detection_and_Forecasting/29931260.

**Funding:** The author(s) received no specific funding for this work.

**Competing interests:** The authors have declared that no competing interests exist.

remains stable; decreasing MASE from 6.9 to 5.3 (US), 12.2 to 8.5 (UK), and 16.0 to 15.2 (Australia), highlighting its robustness over extended periods. These results highlight how separately clustering anomalies allows forecasting models to better capture distinct market behaviors, ensuring more precise and risk-adjusted predictions.

## Introduction

The real estate market remains a cornerstone of the global economy, driving financial stability, urban development and individual wealth [1–4]. As one of the most influential sectors, it plays a pivotal role in shaping macroeconomic conditions, urban planning strategies, and the financial well-being of households and businesses alike [4]. Real estate investments are central to wealth accumulation for individuals and institutional portfolios, while property development contributes to urbanization and the creation of infrastructure essential for economic growth. Accurate forecasting in this domain is critical for a wide range of stakeholders, including investors seeking to maximize returns, developers planning large-scale projects, and policymakers aiming to balance economic growth with risk mitigation. Reliable predictions of real estate market trends enable these groups to navigate uncertainties, identify emerging opportunities, and implement proactive strategies to address market risks.

Despite its importance, the inherent complexity of real estate markets poses significant challenges to reliable forecasting. One of the primary barriers is the presence of anomalies—unexpected and irregular changes in market behavior that disrupt typical patterns [5–9]. Anomalies can arise from a multitude of factors [10], including sudden economic shocks, policy interventions, or unforeseen socio-political developments such as international conflicts, pandemics, or natural disasters. These disruptions introduce volatility and unpredictability, rendering traditional forecasting models ineffective. For instance, a sudden shift in monetary policy, such as changes in interest rates or taxation, can lead to abrupt fluctuations in property demand in specific regions, causing erratic pricing trends and transaction volumes. Failure to account for these anomalies often results in inaccurate forecasts, undermining the ability of stakeholders to make informed decisions.

Given the multifaceted nature of real estate market disruptions, robust anomaly detection methods are useful to improve forecasting reliability. Anomalies, whether temporal or geospatial, are not merely statistical outliers but critical indicators of emerging opportunities and risks. Temporal anomalies, such as sudden spikes or drops in property prices or transaction volumes, may highlight underlying shifts in market dynamics, signaling early signs of investment hotspots or the formation of speculative bubbles. Similarly, geospatial anomalies, such as unexpected price increases in certain suburbs, can indicate localized attractiveness due to new infrastructure developments, policy changes, or other region-specific factors [11–13]. These anomalies provide actionable insights that can guide major investors and developers, such as Mitsubishi Estate Co., Ltd. [14,15] in strategically navigating market complexities, identifying lucrative opportunities, and mitigating potential risks in new real estate investments [16]. However, traditional forecasting approaches often misinterpret or discard anomalies as noise, prioritizing the smoothing of disruptions over the extraction of valuable insights. This approach undermines the predictive accuracy of these models and limits their ability to inform effective strategies [17].

Another significant challenge lies in the lack of integration between geospatial data and trading volume in most existing forecasting methodologies for real estate [18,19]. Real estate

markets are inherently spatial, with location-specific factors such as infrastructure, demographics, and proximity to amenities influencing property values and transaction patterns. At the same time, trading volume provides critical insights into how public decisions shapes market trends, reflecting the collective psychology of buyers, sellers, and investors. The failure to combine these dimensions prevents a holistic understanding of market dynamics, leaving significant gaps in the ability to predict and respond to anomalies.

Addressing these challenges requires a paradigm shift in anomaly detection and forecasting methodologies. This paper proposes a novel graph-based neural network framework, called EGCN, that integrates entropy-based multivariate feature assimilation, a deep graph convolutional network (GCN) based autoencoder to identify and cluster normal and anomalous regions/suburbs and a cluster-specific deep neural time-series forecasting model to enhance predictive accuracy in the real estate market (Fig 1). The framework uses a GCN-based autoencoder in which the encoder component propagates and aggregates spatiotemporal features, capturing both local and global patterns in the data. The decoder component reconstructs the original features from the learned representations, enabling the identification of anomalies as deviations between reconstructed and actual features [20]. Temporal entropy, calculated using Kernel Density Estimation [21], highlights variability in property prices, and transaction volumes, forming the basis for anomaly detection. A graph is constructed where nodes represent suburbs, and edges encode geospatial proximity [22,23] and temporal correlations between the features. The GCN learns meaningful embeddings of the graph that support grouping of suburbs (nodes of the graph) that show similar patterns, i.e., grouping into two distinct clusters: normal and anomalous clusters. These clusters enable the use of distinct tailored deep neural time series forecasting models fitted for each cluster to improve

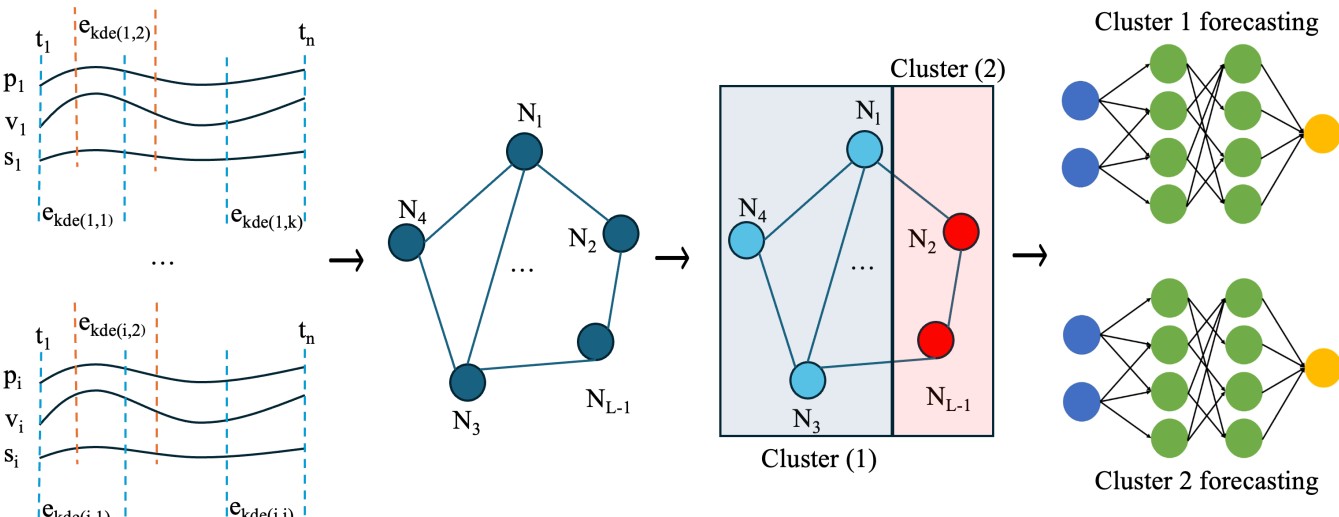

**Fig 1. The proposed framework, EGCN, integrates temporal entropy analysis, graph-based modeling, clustering, and forecasting to detect anomalies and predict future trends in spatiotemporal data**. It begins with multi-dimensional time-series data for $m$ entities, each with features such as price ($p$), volume ($v$) and geospatial data ($s$), sampled at $n$ time points ($t_1, t_2, \dots, t_n$). Entropy values $e_{kde(i,j)}$ are computed for each entity $i$ and time window $j$ using Kernel Density Estimation, capturing temporal variability across features. These entropy values form a matrix representing temporal dynamics. A graph-based autoencoder $G = (V, E)$ is constructed to detect anomaly pattern nodes, where nodes $N_i \in V$ represent entities with features ($e_{kde}$, latitude, longitude), and edges $E_{ij}$ connect nodes based on spatial relationships with a threshold $\tau$. Edges also encode temporal dependencies, such as shared feature correlations or entropy dynamics. Clustering is applied to group the nodes into two clusters, $C_1(non-anomaly)$, $C_2(anomaly)$, grouping entities with similar spatiotemporal behaviors. Each cluster is then independently trained using neural network models, such as Multi-Layer Perceptrons, to predict future trends. This framework effectively combines entropy-based anomaly detection, graph modeling, and cluster-specific forecasting for robust spatiotemporal analysis.

predictive accuracy, compared to a method where all the time series measurements are combined and used as a whole (without clustering) for forecasting. Our evaluation reveals that the cluster-specific fornicating model fitting improves forecasting accuracy. Moreover, by integrating real estate market, and geospatial features, this framework addresses key limitations in existing methods, offering actionable insights for investors, developers, and policymakers.

The key contributions of this study are three fold:

- **Entropy-Based Graph Convolutional Network for Anomaly Detection**: Introducing a novel entropy-based graph convolutional network method that leverages an autoencoder architecture to detect anomalies in graph-structured real estate multi-dimensional time-series data. The method integrates geospatial proximity and temporal trends into its graph representation to identify spatial (suburb-level) and temporal (time-point) resolution anomalies. This approach also provides actionable insights for industrial stakeholders by analyzing features such as property prices, transaction volumes, and geographical locations, enabling the detection of emerging investment opportunities and potential market risks.
- **Clustering suburbs into normal and anomalies that exhibit shared behaviour**: Proposing an approach that groups suburbs or entities into normal and anomalies (two clusters) based on their entropy patterns and graph-based relationships. This process reveals interconnected market behaviors and facilitates a deeper understanding of how localized anomalies propagate through the broader spatial network, offering a valuable tool for regional market analysis.
- **Entropy-based Graph Convolutional Network for Clustering and Forecasting:** Developing advanced cluster-specific forecasting models tailored to predict future trends for each cluster. By focusing on the unique characteristics of each group, these models improve predictive accuracy and relevance compared to traditional global approaches. The EGCN framework demonstrates its ability to outperform baseline methods by accounting for the shared dynamics of clustered nodes, providing more precise and actionable forecasting outcomes for stakeholders.

By addressing the challenges faced by real estate industries—such as forecasting inaccuracies caused by anomalies and underutilized geospatial data, this research offers actionable insights for improving investment strategies. Additionally, the framework's scalability makes it adaptable to other regions and global markets, further underscoring its potential impact. In the following sections, the methodology, experimental results, and practical implications of this approach is explored in detail, demonstrating its efficacy in advancing anomaly detection and forecasting for the real estate industry.

## Related work

The real estate market has long been a focus of research due to its significant impact on global economies and urban development. Accurate forecasting in this domain requires integrating methods from time-series analysis, anomaly detection and machine learning. This section reviews existing approaches and highlights the gaps addressed by the proposed framework.

### Anomaly detection

Anomaly detection has emerged as a crucial task in time-series analysis, especially for identifying disruptions that may signal risks or opportunities in the real estate market. Traditional anomaly detection techniques, such as Z-score analysis and Isolation Forest [24], focus on statistical deviations [25]. Recent advancements have introduced machine learning-based

approaches, including Autoencoders [26] and Variational Autoencoders [27], to model normal patterns and identify deviations. However, these methods are limited in their ability to combine temporal, and spatial factors [28,29]. Existing methods often misinterpret such anomalies as noise, missing critical insights into localized market dynamics.

## Graph-based models for spatiotemporal data

Graph-based models, such as Graph Neural Networks (GNNs) [30], GCNs [31], have shown significant promise in capturing spatiotemporal dependencies in various domains, including transportation, climate analysis, and social networks [11,12]. By representing entities as nodes and relationships as edges, graph-based models can effectively model complex interactions between spatially distributed features. However, the application of graph-based models in real estate markets remains limited, with most studies focusing on either spatial or temporal factors in isolation. Existing models rarely integrate entropy-based anomaly detection into graph representations, leaving significant gaps in addressing the multidimensional nature of real estate markets [32].

## Clustering and cluster-specific forecasting

Clustering techniques, such as K-Means [33], Dynamic Time Warping (DTW) [34], and Weighted Dynamic Time Warping (WDTW) [35], have been widely used to group entities with similar behaviors, enabling localized insights and targeted analysis. In real estate, clustering can reveal regional trends, such as neighborhoods experiencing rapid growth or decline [7,36–38]. However, most clustering approaches fail to incorporate anomaly-driven patterns, which are critical for understanding market disruptions. Furthermore, traditional forecasting methods often treat clusters as static entities, ignoring the dynamic interactions within and across clusters. Advanced forecasting methods, such as recurrent neural networks (RNNs) [39] and transformers-based models [40], offer potential for improving cluster-specific predictions but are rarely tailored to account for shared anomaly-driven features within clusters.

## Multivariate time series forecasting

Recent advancements in multivariate time-series forecasting have demonstrated the effectiveness of MLP-based models in enhancing prediction accuracy and reliability within financial markets. Long short-term memory (LSTM) models, known for their capability to capture long-term dependencies in sequential data, have been applied successfully in forecasting stock market trends [41] and gold prices [42]. The neural hierarchical interpolation for time series (N-HiTS) model has further shown its utility in efficiently identifying trends and patterns in multivariate financial datasets, particularly in forecasting financial indices [43–46]. Similarly, time series mixer (TSMixer), a model tailored to address the inherent volatility of financial markets, has achieved strong performance in predicting stock prices and exchange rates, highlighting its role in financial time-series analysis [47,48]. Meanwhile, Transformers [49], initially designed for natural language processing, have emerged as powerful tools for time-series forecasting, celling in capturing complex temporal dependencies and long-range patterns in financial data.

## Methodology

This section presents the proposed framework, EGCN, for anomaly detection and forecasting in the real estate market. The framework integrates temporal entropy analysis, graph-based encoder modeling, anomalies detection, and forecasting to provide actionable insights.

It consists of four main stages: temporal entropy computation, graph construction, anomaly clustering, and cluster-specific forecasting. The framework leverages multi-dimensional data, including property prices, and transaction volumes to identify anomalies and predict market trends.

## Temporal entropy-based analysis

Let $X = \{x_1, x_2, \ldots, x_T\}$ be the multivariate time-series data for $n$ entities, where $x_t = (x_{t1}, x_{t2}, \ldots, x_{td})$ is a $d$-dimensional feature vector at time $t$. The first step involves applying a sliding window of size $W$ to $X$ to extract overlapping segments of the time series:

$$X_w = \begin{bmatrix} x_{t1} & \cdots & x_{tw} \\ \vdots & \ddots & \vdots \\ x_{(t+W-1)1} & \cdots & x_{(t+W-1)w} \end{bmatrix}. \tag{1}$$

where $X_w$ represents a windowed segment of the data. The output $X_w$ is then used to compute the entropy of each segment to quantify variability.

**Kernel density estimation computation.** For each window $X_w$, we estimate its probability distribution using Kernel Density Estimation (KDE) [50]:

$$\hat{f}_w(x) = \frac{1}{N} \sum_{i=1}^{N} K\left(\frac{x - x_i}{h}\right), \tag{2}$$

where $K$ is the kernel function (e.g., Gaussian), $h$ is the bandwidth, and $N$ is the number of points in the window. The KDE-estimated distribution is then used to serve as the input for the next step, where temporal divergence between distributions is measured.

Kernel Density Estimation (KDE) is highly appropriate for the proposed framework due to its ability to estimate the underlying probability density functions of complex and noisy data without assuming any specific parametric form [51,52]. This non-parametric approach is crucial for real estate forecasting and anomaly detection, where the data distributions of features such as property prices, and transaction volumes are often irregular and multimodal. By using KDE, the framework captures the nuanced variations in these distributions across temporal windows, enabling the identification of subtle shifts or anomalies that traditional methods may overlook. Additionally, KDE's flexibility allows it to handle multivariate data effectively, making it ideal for integrating spatial, and temporal dimensions inherent in real estate markets. This capability ensures that KDE not only enhances the accuracy of anomaly detection but also provides a robust foundation for predicting future trends based on probabilistic patterns, aligning with the framework's goals for dynamic market analysis.

**Jensen-Shannon Divergence (JSD).** To capture changes between successive sliding windows, the Jensen-Shannon Divergence (JSD) [53] is computed:

$$JSD(w, w+1) = \frac{1}{2}\left[D_{KL}(\hat{f}_w \| \hat{f}_{w+1}) + D_{KL}(\hat{f}_{w+1} \| \hat{f}_w)\right], \tag{3}$$

where $D_{KL}$ is the Kullback-Leibler divergence. The sequence of JSD values $\{JSD(w, w+1)\}$ identifies significant temporal anomalies, forming the basis for graph construction.

The Jensen-Shannon Divergence (JSD), a symmetric and smoothed extension of Kullback-Leibler Divergence (KLD), is highly suitable for the proposed framework due to its robust ability to quantify the similarity between probability distributions by integrating KLD and

Shannon entropy [53,54]. Unlike KLD, JSD is symmetric, ensuring that the order of distributions does not affect the divergence value, which is essential for clustering and anomaly detection tasks requiring consistent similarity measurements. Moreover, the incorporation of Shannon entropy introduces a smoothing effect, reducing sensitivity to noise and small variations, making it particularly effective for analyzing real-world multivariate time-series data. Within our framework, JSD facilitates the comparison of probability distributions over temporal windows, enabling the detection of significant deviations in features such as prices, and volumes while remaining robust to outliers. Its capability to capture both temporal and distributional differences makes JSD a powerful metric for anomaly detection and clustering, aligning seamlessly with the objectives of the proposed method.

To quantify temporal variability, we adopt KDE combined with JSD rather than alternative measures such as Shannon or Rényi entropy. KDE is non-parametric and well-suited for real estate data, which often exhibit multimodal and heavy-tailed distributions due to abrupt price shifts and irregular trading volumes, while JSD provides a symmetric and bounded divergence measure that incorporates Shannon entropy to smooth fluctuations and reduce sensitivity to noise. In contrast, Shannon entropy is highly dependent on discretization and may underperform when distributions are multimodal, and Rényi entropy requires parametric tuning that can lead to instability across heterogeneous datasets [55]. By applying JSD to KDE-estimated distributions across consecutive time windows, our framework captures both distributional and temporal changes, enabling robust detection of subtle market disruptions. The entropy and divergence values form the foundation for constructing a spatiotemporal graph representation.

## Graph construction

The entropy and JSD values are used to construct a graph $G = (V, E)$, where nodes represent entities and edges capture relationships based on spatial and temporal features [30].

**Node features.** Each node $N_i \in V$ is associated with a feature vector derived from JSD and spatial coordinates:

$$N_i = \{\text{Entropy}, \text{Latitude}, \text{Longitude}\}. \tag{4}$$

These node features encode both temporal variability and spatial characteristics.

**Edge construction.** Edges $E_{ij}$ between nodes $N_i$ and $N_j$ are defined based on spatial proximity and temporal similarity:

- **Spatial Proximity:** Edges are created if the geospatial distance between nodes satisfies Haversine distances:

$$d(i,j) = 2R \arcsin \left( \sqrt{\sin^2 \left( \frac{\phi_j - \phi_i}{2} \right)} \right.$$
$$\left. + \cos(\phi_i) \cos(\phi_j) \sin^2 \left( \frac{\lambda_j - \lambda_i}{2} \right) \right) \tag{5}$$

where $\phi$ and $\lambda$ are latitudes and longitudes, $R$ is Earth's radius.

- **Temporal Similarity:** Edge weights are proportional to the correlation of entropy values between nodes over time. The final edge weight is defined as:

$$w_{ij} = \alpha \, S_{ij}^{\text{spatial}} + (1 - \alpha) \, S_{ij}^{\text{temporal}}, \tag{6}$$

where $\alpha \in [0, 1]$ is a balancing coefficient. This design ensures that both geographic closeness and similarity in temporal variability jointly determine the strength of connections in the graph. In our experiments, $\alpha$ was set to 0.5 to give equal importance to spatial and temporal features, but this parameter can be tuned to emphasize one dimension if desired. By combining spatial and temporal similarity into a single edge weight, the constructed graph better reflects the multidimensional relationships that drive anomalies in real estate markets.

The resulting graph structure $G$ serves as input for the next step. Once the graph is defined, we employ a GCN to learn embeddings that capture these spatiotemporal relationships.

## Graph convolutional network

Entropy values obtained from previous steps are utilized in a GCN-based autoencoder to construct the graph structure, facilitating the detection of spatiotemporal anomalies. The GCN in this framework employs an encoder-decoder architecture tailored to the real estate market case study. The encoder propagates and aggregates entropy values from neighboring suburbs, learning low-dimensional node embeddings that capture both local market behaviors and global patterns within the graph. The decoder reconstructs the original entropy values, ensuring that the learned embeddings preserve essential spatiotemporal dynamics. Anomalies, such as irregular trends in property prices, and transaction volumes are identified by comparing the reconstructed entropy values with the original data, with significant deviations highlighting potential spatiotemporal anomalies. The process follows these key steps:

- **Feature Propagation and Aggregation:** Node features are propagated and aggregated using edge connections, allowing nodes to integrate information from their neighbors. Each node $N_i$ updates its representation by combining its own features with those of its neighbors, weighted by the edges $E_{ij}$.
- **Layer-wise Node Embedding Updates:** At each layer $l$, the embedding of node $i$ is updated as:

$$h_i^{(l+1)} = \sigma \left( \sum_{j \in \mathcal{N}(i)} \frac{1}{\sqrt{\deg(i) \cdot \deg(j)}} W^{(l)} h_j^{(l)} \right), \tag{7}$$

where: $h_i^{(l)}$: Embedding of node $i$ at layer $l$; $\mathcal{N}(i)$: Neighbors of node $i$; $\deg(i)$: Degree of node $i$; $W^{(l)}$: Trainable weight matrix for layer $l$; $\sigma(\cdot)$: Non-linear activation function (e.g., ReLU).
- **Final Embeddings:** After propagating through multiple layers, the final embeddings capture both local patterns (relationships between neighboring nodes) and global patterns (graph-wide trends).

These embeddings allow us to reconstruct node features and evaluate deviations, which provides the basis for anomaly detection.

## Identification of anomalous nodes

Anomaly detection begins by calculating the reconstruction error for each suburb, comparing its original entropy values, derived from $w$ sliding windows of the dataset, with the reconstructed values produced by the trained GCN. The reconstruction errors across all features of a node are averaged to compute an aggregate error, excluding spatial attributes like latitude and longitude. A threshold is then determined based on the distribution of aggregate

errors across all nodes. Nodes with aggregate errors exceeding this threshold are identified as anomalous, highlighting significant deviations from normal patterns. The detailed process is shown below:

- **Thresholding:** Nodes $N_i$ and $N_j$ with $d(i,j) > $ *Threshold* are flagged as anomalous due to spatial isolation or irregularity.

$$\text{Threshold} = \mu_{\text{aggregate\_error}} + k \cdot \sigma_{\text{aggregate\_error}}, \tag{8}$$

$$\text{Aggregate Error}_i = \frac{1}{n} \sum_{j=1}^{n} \text{Reconstruction Error}_{ij}. \tag{9}$$

where $\mu_{\text{aggregate\_error}}$ represents the mean of the aggregate reconstruction errors across all nodes, $\sigma_{\text{aggregate\_error}}$ is the standard deviation of the aggregate reconstruction errors, and $k$ is a threshold multiplier that controls the sensitivity of anomaly detection. Under a Gaussian assumption, $\mu + 1.5\sigma$ corresponds to a one-sided tail probability of approximately 6.7%, matching our operational aim to highlight only the most atypical regions [56–58]. We favor this fixed, a priori choice to prevent test-set–driven threshold optimization. Future work may consider data-driven calibration (e.g., targeting a desired false-positive rate or using validation folds) where appropriate.

- **Anomalous Grouping:** Nodes are assigned into two clusters, including flagged nodes (cluster of anomalies), and normal nodes (the rest).

$$\text{Aggregate Error}_i > \text{Threshold} \tag{10}$$

The graph is then segmented into clusters based on spatial relationships and feature similarities. Nodes flagged as anomalous are grouped into a distinct cluster, separating them from regular patterns, while the remaining nodes are organized into regular clusters that represent typical behaviors. This methodology is highly effective, practical, and directly relevant for decision-making due to the following key reasons:

- **Integration of Spatiotemporal Features:** Combines entropy, latitude, and longitude to analyze spatial and temporal patterns effectively.
- **Scalable Framework:** GCN enables efficient processing of large datasets.
- **Business Relevance:** Domain-specific knowledge ensures the approach is aligned with real-world applications.

After identifying anomalous nodes, we separate the data into clusters, which then serve as distinct inputs for cluster-specific forecasting models. Besides, it is important to note that anomalies in our framework are identified at the location level (e.g., suburb or city) rather than at specific location–time pairs. The anomaly score for each node is computed from the aggregate reconstruction error of its entropy series across the full observation window. While this means the anomaly flag is assigned to the node as a whole, temporal dynamics are not ignored: windowed entropy combined with Jensen–Shannon Divergence ensures that short-term fluctuations and recurrent deviations contribute to the reconstruction error. Thus, transient shocks influence anomaly scores indirectly, but the final anomaly designation reflects persistent or structural irregularities at the regional level.

## Cluster-specific forecasting models

Following training, each node's anomaly score is defined as the norm of the reconstruction residual produced by the GCN autoencoder. We then perform a *threshold-based binary partition* using a robust z-score rule: nodes with scores exceeding $\mu_{\text{agg}} + 1.5\,\sigma_{\text{agg}}$ are labeled *abnormal*, and all others *normal*. This yields the two groups reported in our results without invoking an additional clustering algorithm such as K-means, Spectral Clustering, or GMM. To check robustness, we evaluate the partition under a grid of hyperparameters (window size, hidden channels, learning rate, distance cutoff, and training epochs) and find that the identity of abnormal nodes is largely consistent across settings, indicating stability of the decision rule. Using the above GCN-based anomalies detection, the suburb have 2 clusters. Each cluster $C_k$ is treated as an independent unit for forecasting:

$$\hat{X}_{t+1} = f(X_{t-W:t}, \text{Cluster Features}) \tag{11}$$

where $f$ is the forecasting model, $X$ is the historical data, and $W$ is the time window.

Forecasting methods used in this study include N-HiTS, Transformer, and TSMixer. Input features include historical prices, and transaction volumes.

N-HiTS [43], an extension of N-BEATS [59], improves forecasting accuracy while reducing computational complexity [43,60,61]. However, N-HiTS encounters difficulties in maintaining high accuracy with intricate dynamic patterns in multivariate data. It employs a hierarchical approach for time series forecasting, using blocks composed of multi-layer perceptrons (MLPs) to predict coefficients [43]. To enhance long-term forecasting, MaxPool layers with kernel size $k_\ell$ are applied to focus on specific scale components. The input to block $\ell$ is given by:

$$\mathbf{y}_{t-L:t,\ell}^{(p)} = \text{MaxPool}\left(\mathbf{y}_{t-L:t,\ell}, k_\ell\right) \tag{12}$$

where $\mathbf{y}_{t-L:t,\ell}$ represents the input data from time steps $t-L$ to $t$ for block $\ell$, and $\mathbf{y}_{t-L:t,\ell}^{(p)}$ is the output of MaxPool, extracting optimized scale components for improved forecasting. These coefficients are used to generate the backcast, $\tilde{\mathbf{y}}_{t-L:t,\ell}$, and the forecast, $\hat{\mathbf{y}}_{t+1:t+H,\ell}$, which are the respective outputs of the block.

Transformers [49] uses a self-attention mechanism to model dependencies in sequences:

$$\text{Attention}(Q, K, V) = \text{softmax}\left(\frac{QK^T}{\sqrt{d_k}}\right) V \tag{13}$$

where $Q$, $K$, $V$ are the query, key, and value matrices, and $d_k$ is the dimensionality of the key vectors.

TSMixer [47] is a novel neural network architecture designed to capture intricate patterns in time series data by stacking multiple multi-layer perceptrons (MLPs). The architecture consists of several components: an input layer, a mixer layer comprising multiple MLPs, an aggregation layer, and a final output layer. The forecasted value is computed as follows:

$$\hat{y}^t = \sigma\left(\text{concat}\left(\text{MLP}_1(x_t), \ldots, \text{MLP}_n(x_t)\right) W\right) \tag{14}$$

where $x_t$ represents the input data at time step $t$, $\text{MLP}_i(x_t)$ is the output of the $i$-th MLP, concat denotes the concatenation of the outputs from all MLPs, $W$ is a learnable weight matrix, and $\sigma$ is the activation function.

## Original contributions compared to existing methods

Most existing graph-based anomaly detection models in spatiotemporal domains primarily focus on identifying anomalous nodes or regions using raw features (e.g., prices or volumes) aggregated through graph structures. While effective for detection, these models do not explicitly incorporate temporal entropy dynamics or use anomalies to enhance downstream forecasting. In contrast, our proposed framework introduces several key innovations:

1. **Entropy-driven representation:** Instead of relying only on raw time-series values, EGCN computes temporal entropy using Kernel Density Estimation (KDE) and Jensen–Shannon Divergence (JSD). This entropy-based formulation provides a distributional view of variability in prices and transaction volumes, making anomalies more robustly detectable than with simple statistical or embedding methods.
2. **Integration of geospatial and temporal similarity:** EGCN explicitly constructs graph edges using a combination of Haversine distance (spatial proximity) and temporal correlation (similarity of entropy dynamics). This dual-edge design ensures that both location-based and temporal behavioral relationships are preserved in the graph, improving anomaly detection compared to baselines that consider only one dimension.
3. **Anomaly-aware clustering and forecasting:** Whereas baseline GNN-based approaches terminate at anomaly detection, EGCN extends further by clustering anomalous and normal regions separately and applying cluster-specific forecasting models. This enables forecasting models to specialize for distinct market behaviors, reducing error across short, mid, and long-term horizons.
4. **Cross-country validation:** Unlike prior GNN-based methods, which are often evaluated within a single market, EGCN is validated on datasets from three countries (Australia, UK, and USA). Additional cross-country experiments demonstrate its ability to generalize cluster-specific forecasting across distinct housing systems.

Together, these contributions establish EGCN as the first framework to jointly integrate entropy-based temporal variability, geospatial proximity, and cluster-specific forecasting within a unified graph convolutional architecture. This positions EGCN as a significant advancement beyond existing methods that focus solely on anomaly detection.

## Experimental setup

The experiments are designed to assess the anomaly detection, clustering and forecasting performance of the EGCN. Comparative analyses are performed against state-of-the-art graph-based methods, demonstrating the superiority of our approach in capturing complex spatio-temporal dependencies and delivering actionable insights for urban analytics.

### Data preparation

We utilize three datasets that include time-series real estate data, namely price and volume, for various regions from Australia, the United States of America, and the United Kingdom from 01/2003 to 06/2024, along with their geospatial coordinates. It is important to note that anomaly detection is conducted as an unsupervised diagnostic analysis on the entire dataset from 2003 to 2024 to identify irregular spatiotemporal patterns. This step does not involve prediction and therefore can leverage the full observation window. By contrast, forecasting strictly follows a temporal train/test split: models are trained on the pre-COVID period from 2003 to 2019 and evaluated on the post-COVID period from 2020 to 2024,

widely regarded as one of the most unstable periods in the past 30 years [62]. This separation ensures that predictive performance is assessed without any exposure to future data. This temporal division allows us to evaluate the robustness and adaptability of the proposed method in forecasting real estate dynamics under highly volatile and unpredictable conditions. All code and data used in this study are openly available at the following DOI link: 10.6084/m9.figshare.29931260.

- **Australia:** Historical real estate data covering 1,695 suburbs were obtained from Australian Property Monitors via the AURIN portal (data.aurin.org.au).
- **United States:** Real estate data from 301 major cities were sourced from Zillow (zillow.com), the nation's largest online real estate platform. The dataset and maps focus on the contiguous 48 states; Alaska and Hawaii were excluded due to limited Zillow coverage during 2003–2024 and for visualization clarity. Zillow's geographic coverage was more limited before 2007, introducing sparsity in some localities. To ensure data quality, suburbs with persistent missing values were excluded (removals less than 5% of the total dataset), and all series were winsorized at the 1st and 99th percentiles to mitigate the influence of extreme early observations. These steps ensured consistent training quality across the 2003–2019 sample period.
- **United Kingdom:** Property transaction data were obtained from HM Land Registry (https://www.gov.uk/government/organisations/land-registry), covering 1053 suburbs in England, Wales, and Northern Ireland. Scotland was excluded because comparable suburb-level transaction data were not consistently available across the study period, reflecting its distinct legal and housing data systems.

## Baseline comparisons

To evaluate the effectiveness of our proposed approach, we compare its performance against several well-established methods for detecting anomalies in real estate data. Each method is based on **Graph Neural Networks (GNNs)** and is designed to identify regions where property prices or transaction volumes exhibit unusual trends. Below, we describe the baseline methods used in this study.

- **Graph Autoencoder (GAE):** The GAE [63] is an unsupervised learning model that identifies patterns in real estate data by compressing and reconstructing information about different regions. If a region deviates significantly from historical trends, the model struggles to reconstruct its features accurately, leading to a high reconstruction error. This makes GAE useful for detecting suburbs or cities where property market trends behave abnormally.
- **Graph Attention Network (GAT):** GAT [64,65] improves anomaly detection by learning the relative importance of neighboring regions. Unlike conventional methods, which treat all regional connections equally, GAT assigns different importance weights to areas based on their influence on each other. This is particularly useful in real estate markets, where some locations have stronger economic connections than others.
- **Graph Sample and Aggregate (GraphSAGE):** GraphSAGE [66] is designed for large datasets, sampling, and aggregating information from neighboring regions. This enables the model to detect anomalies in real estate markets efficiently, especially when a small number of suburbs experience sudden price shifts that may indicate economic disruptions or emerging trends.
- **Graph Convolutional Network (GCN):** GCN [67] learns property price and transaction patterns by aggregating data from connected regions. It helps identify suburbs or cities that

do not follow expected market trends. This method is particularly useful for detecting spatial anomalies, such as a neighborhood experiencing rapid property price increases while surrounding areas remain stable.
- **Graph Temporal Attention (GTA)**: GTA [68] extends traditional methods by incorporating time-series data. It reshapes real estate data into sequences and applies attention mechanisms to understand how past trends influence present behavior. By analyzing patterns over time, GTA can detect anomalies that emerge gradually, making it valuable for predicting long-term market sustainability.
- **Multi-Temporal Graph Neural Network (MTGNN)**: MTGNN [69] combines both spatial and temporal information to analyze real estate trends. It models how different regions are connected while also tracking how property market patterns evolve. This enables the detection of anomalies such as a city experiencing rapid price fluctuations while nearby suburbs remain unchanged.

### Evaluation and performance metrics

Each baseline method is trained using historical real estate data, and anomalies are identified based on reconstruction errors. To evaluate the effectiveness of our approach, we assess the models using the following metrics:

- **Number of detected anomalies:** The total count of regions (suburbs, cities, or other areas) identified as exhibiting unusual property market behavior.
- **Impact on forecasting accuracy:** After detecting anomalies, we divide regions into two groups: normal and anomalous regions. We then apply forecasting methods to each group separately, demonstrating that accounting for anomalies improves forecasting accuracy. This validates the effectiveness of our anomaly detection in enhancing real estate market predictions. Evaluation metrics used in this study are mean absolute scaled error (MASE) [70], mean squared error (MSE) [71], and mean absolute error (MAE) [71].
- **Validation using domain knowledge:** To ensure the reliability of detected anomalies, we cross-check the results with real-world real estate trends, historical market events, and expert insights. This additional validation step helps confirm whether the identified anomalies align with known economic shifts, policy changes, or urban development patterns.

By benchmarking against these models, we ensure a comprehensive evaluation of our approach in identifying unusual patterns in property prices and transaction volumes. This comparison helps policymakers, urban planners, and real estate professionals make data-driven decisions based on detected market anomalies.

## Results

This section presents the findings of our study, evaluating the effectiveness of our anomaly detection method and its impact on forecasting accuracy. We compare our results with baseline models and validate them using real estate market trends and domain knowledge.

### Real estate anomaly detection

In Fig 2, the anomaly detection results reveal significant differences across models in identifying unusual real estate market behavior in Australia. The Graph Autoencoder (GAE) and GCN detect a similar number of anomalies (107–108), while GraphSAGE and GTA show slightly higher counts (119-129), suggesting that spatial and temporal information enhances

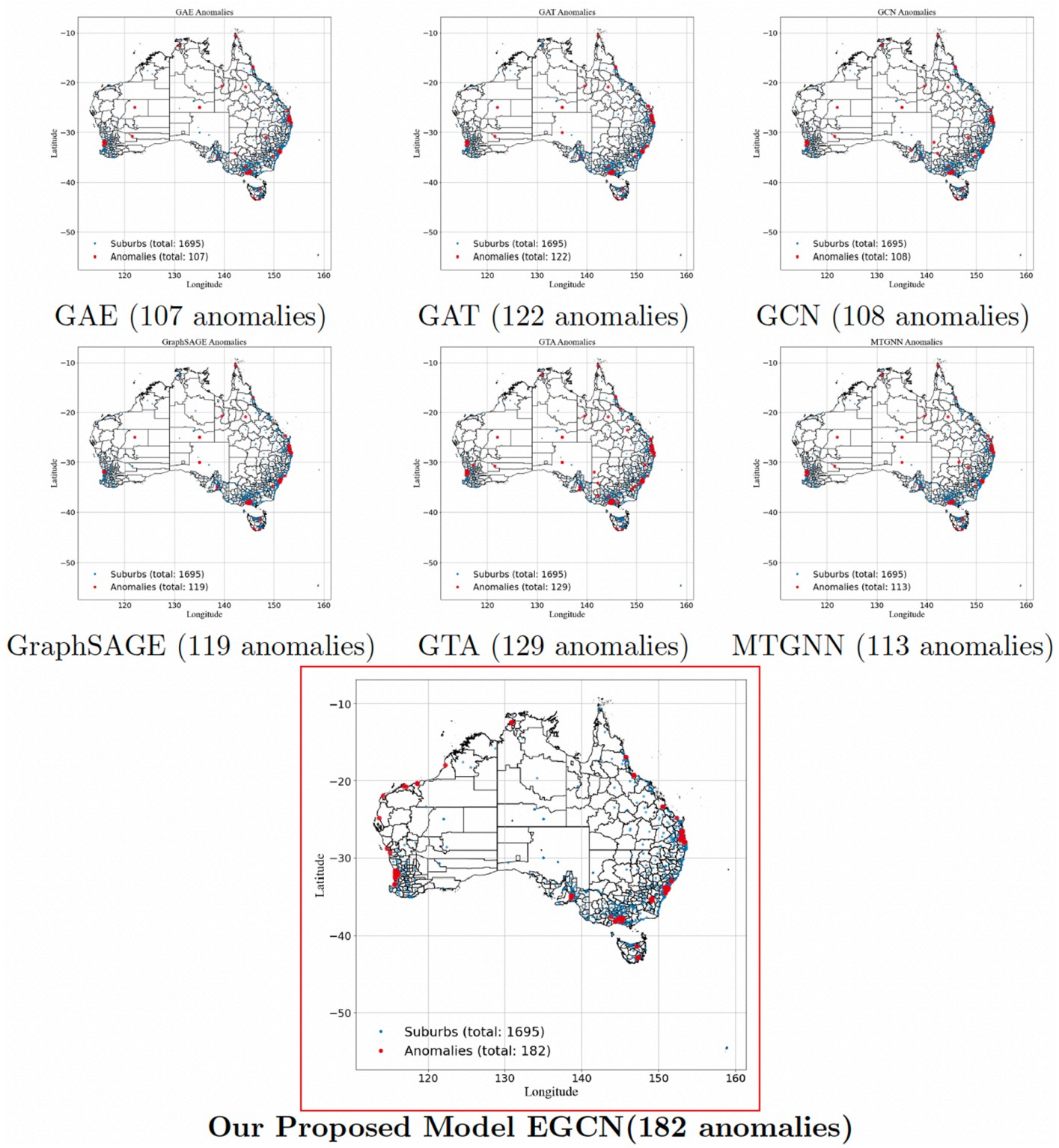

**Fig 2. Comparison of detected anomalies across different models on Australian real estate data.** Reprinted from gadm.org under a CC BY license, with permission from Global Administrative Areas, original copyright 2018–2022.

detection. The Multi-Temporal GNN (MTGNN) further improves results by capturing time-dependent trends, detecting 113 anomalies. The proposed model outperforms all others with 182 detected anomalies, indicating superior sensitivity to market fluctuations. Spatially, anomalies are concentrated in major metropolitan areas such as Sydney, Melbourne, and Brisbane, while regional hubs like Geelong and the Gold Coast also exhibit irregular trends, likely due to market growth or instability. Some detected anomalies in rural areas may indicate unexpected investment shifts. The results derived from the Australian dataset closely reflect the recommendations presented in the *State of the Housing System 2024 Report* [72], highlighting their practical relevance. Validation against real-world trends confirms that the proposed model aligns with known economic changes and housing market patterns. These findings suggest that anomaly detection significantly enhances forecasting accuracy, enabling policymakers, urban planners, and investors to make informed decisions regarding real estate market dynamics.

In Fig 3, the anomaly detection results for the UK real estate market reveal distinct variations across models, with GAE, GCN, and MTGNN detecting the lowest number of anomalies (46-47), suggesting limited sensitivity to subtle market deviations. GraphSAGE and GTA identify slightly more anomalies (52-55), indicating that neighborhood aggregation and temporal awareness improve detection. The proposed model significantly outperforms all others, detecting 117 anomalies, highlighting its superior ability to capture market fluctuations. Spatially, anomalies are concentrated in London and Southeast England, where property prices are highly volatile, while emerging anomalies in the North and Midlands suggest shifting market trends. The findings align with economic shifts, Brexit-related housing uncertainties, and post-pandemic real estate dynamics, as shown in *UK House Price Index summary: December 2024* [73], confirming the model's validity. By distinguishing anomalous vs. normal regions, the results enhance forecasting accuracy, supporting policymakers in identifying housing risks, investors in making data-driven decisions, and urban planners in anticipating future infrastructure needs. The proposed model's increased sensitivity makes it a valuable tool for real estate market analysis and decision-making.

In Fig 4, the anomaly detection results for the US real estate market highlight significant variations across different graph-based models. GAE, GCN, and GAT detect the lowest number of anomalies (12-15), suggesting limited sensitivity to market fluctuations. GraphSAGE and GTA perform slightly better, detecting 16 anomalies, while MTGNN captures 21 anomalies, benefiting from its ability to integrate spatial and temporal dependencies. The proposed model significantly outperforms all others, detecting 34 anomalies, demonstrating a stronger ability to identify unusual market trends. Spatially, anomalies are concentrated in California, Washington, Texas, and the East Coast, which are known for volatile real estate conditions due to high demand, economic shifts, and migration patterns. The detection of anomalies in midwestern and southern states, typically considered stable markets, indicates potential emerging trends or market corrections. These results align with post-pandemic housing shifts and inflation-driven property value fluctuations, reinforcing the effectiveness of anomaly detection in improving real estate forecasting. The findings from the U.S. dataset align closely with the patterns and guidance presented in the *America's Rental Housing 2024 Report* [74], emphasizing their practical importance and applicability to real-world real estate market trends. By distinguishing anomalous from normal regions, these insights can guide investors, policymakers, and urban planners in making data-driven decisions for risk assessment and market intervention.

The anomalies detected by EGCN across Australia, the United Kingdom, and the United States correspond closely with major, well-documented disruptions in real estate markets, supporting the external validity of our approach. In Australia, dense anomaly clusters around

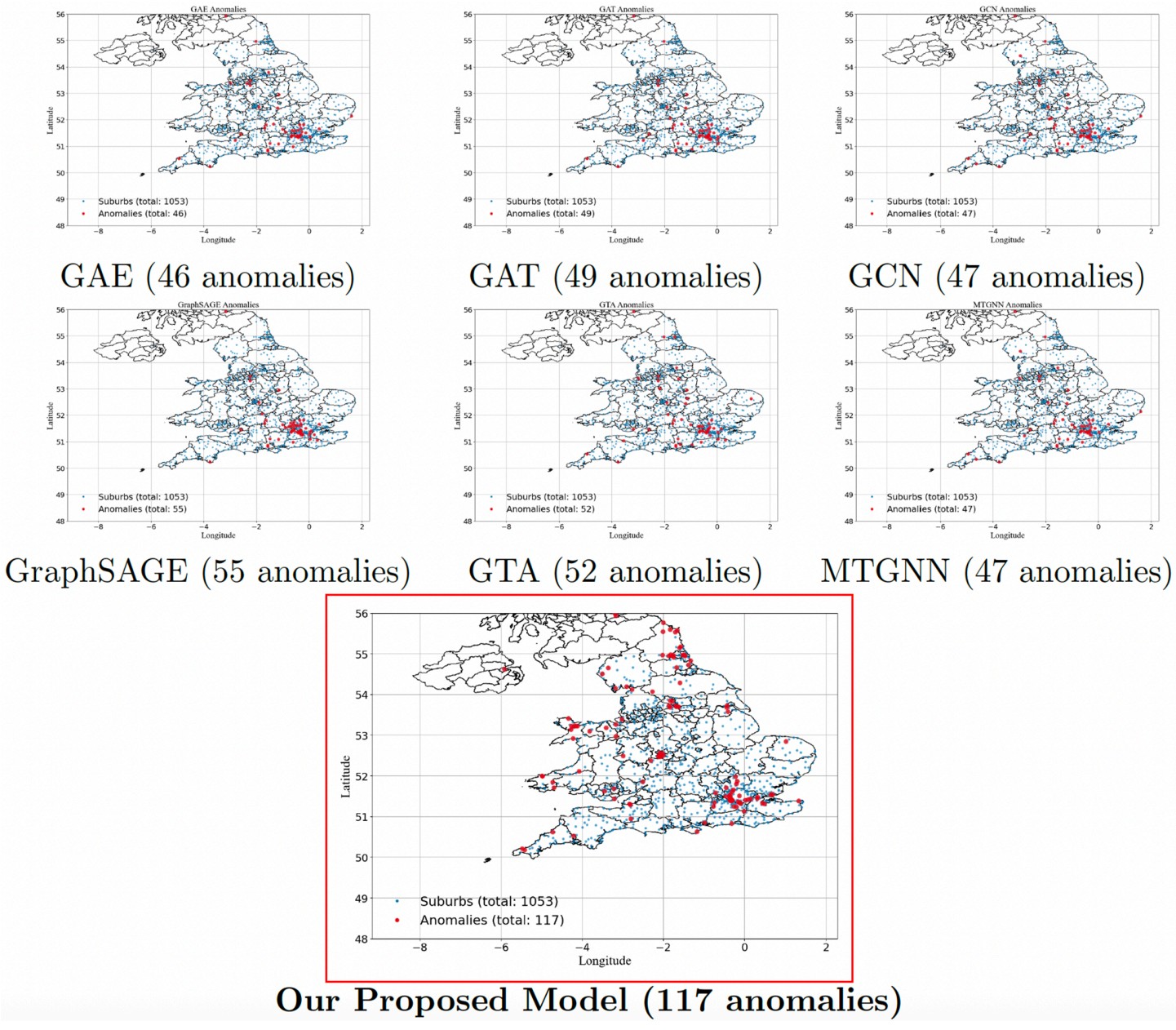

**Fig 3. Comparison of detected anomalies across different models in the UK real estate market.** Reprinted from gadm.org under a CC BY license, with permission from Global Administrative Areas, original copyright 2018–2022.

Sydney and Melbourne during 2020–2021 align with the COVID-19 housing surge, which was fueled by record-low interest rates (0.10% cash rate), mortgage repayment holidays, and government incentives such as the HomeBuilder grant, all of which triggered sharp increases in dwelling prices and transaction volumes [75,76]. In the United Kingdom, anomaly signals are concentrated in London and the South East during 2008–2009, consistent with the global financial crisis, when house prices fell by more than 15% in a single year due to the collapse of Northern Rock and widespread credit tightening [77]. In the United States, anomalies

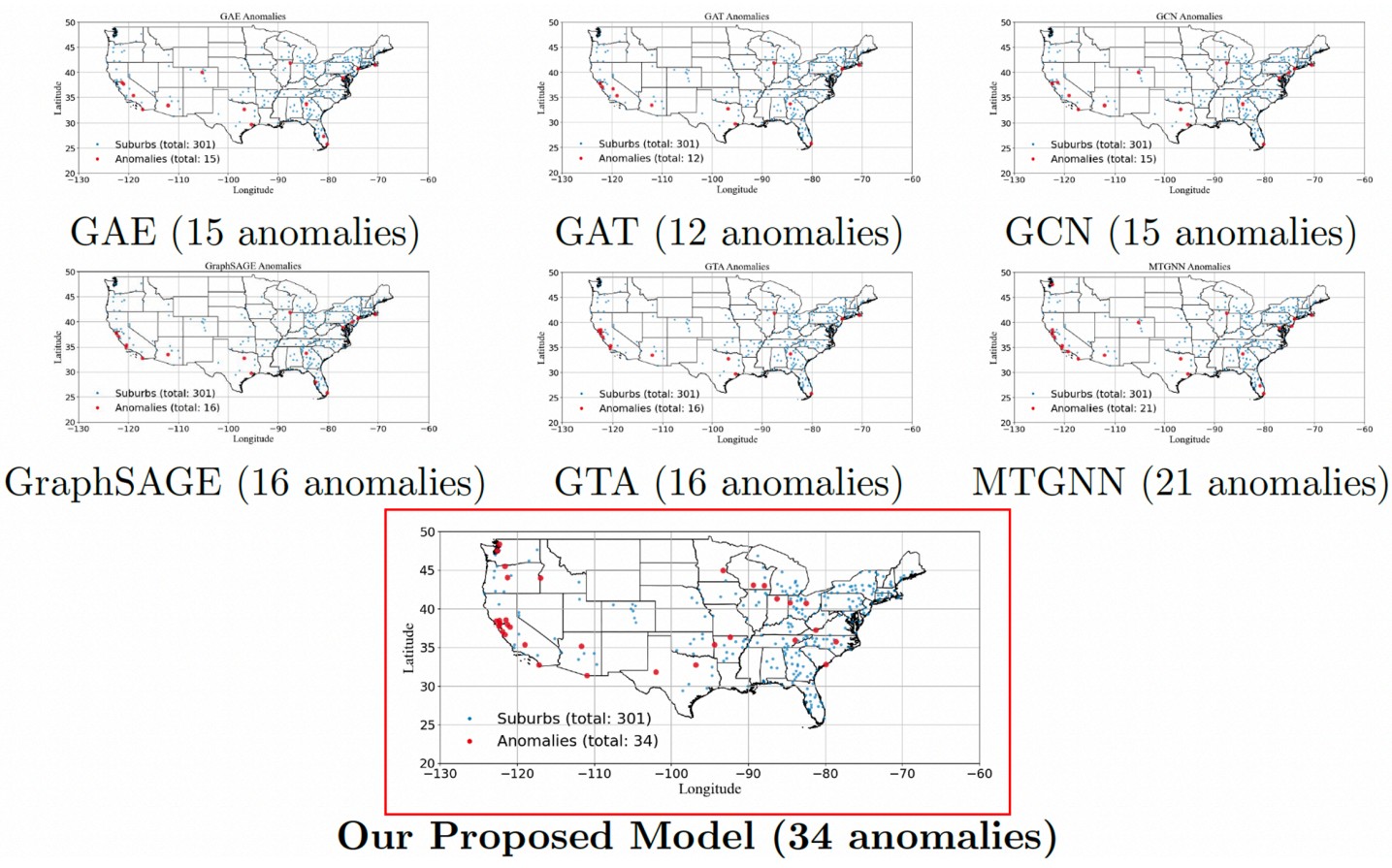

**Fig 4. Comparison of detected anomalies across different models in the US real estate market.** Reprinted from gadm.org under a CC BY license, with permission from Global Administrative Areas, original copyright 2018–2022.

detected in California, Nevada, Arizona, and Florida closely match the epicenters of the 2007–2009 subprime mortgage crisis, where excessive leverage, speculative building, and foreclosure waves produced localized collapses in housing markets [78,79]. Additionally, a second set of anomalies appears in 2020 around major metropolitan areas such as New York, San Francisco, and Seattle, which aligns with COVID-induced migration patterns (urban-to-suburban shifts) and supply-side constraints documented by Zillow and the U.S. Census Bureau [80,81]. The fact that these detected anomalies coincide with distinct macroeconomic shocks across different geographies and time periods suggests that EGCN is not simply flagging statistical outliers but is instead capturing genuine, structurally meaningful disruptions in housing markets.

To ensure that the anomalies identified by EGCN are not artifacts of reconstruction error sensitivity, we conducted placebo tests in which node labels were permuted and temporal sequences shuffled. For each dataset, the procedure was repeated over 50 independent randomizations, and the average number of detected anomalies was recorded. EGCN was then retrained and evaluated on these randomized datasets. Table 1 reports the mean anomaly counts from placebo runs compared with the anomalies detected in the original structured datasets.

**Table 1.** **Average anomalies detected by EGCN across 50 placebo runs compared to anomalies detected in the original datasets.**

| Dataset | Mean placebo anomalies | Original EGCN |
|---|---|---|
| Australia (AUS) | 1 | 182 |
| United Kingdom (UK) | 49 | 117 |
| United States (USA) | 15 | 34 |

In all three markets, anomaly counts under placebo randomization were substantially lower than those obtained from the original data. The effect was most pronounced in Australia, where randomization virtually eliminated anomalies (mean of 1 compared to 182). For the UK and USA, placebo anomaly counts were less than half of those observed in the structured data (49 vs. 117 and 15 vs. 34, respectively).

To further quantify these contrasts, we compared reconstruction error distributions between original and placebo runs using Welch's t-test. In every case, differences were statistically significant ($p < 0.00001$). These results demonstrate that EGCN anomalies are not artifacts of reconstruction error sensitivity, but instead arise from meaningful spatiotemporal dependencies in the housing markets.

## Real estate forecasting

The following tables present the forecasting performance of different models (N-HiTS, TSMixer, and Transformers) across short-term (12 months), mid-term (24 months), and long-term (48 months) horizons for the UK, US, and Australian real estate markets. Each table compares the impact of different anomaly detection methods, where anomalies are clustered separately from normal regions before forecasting. The evaluation metrics include MASE, MSE, and MAE. The best-performing results are presented in bold and underlined, while the second-best results are shown in bold only. These results demonstrate that EGCN achieves the lowest forecasting errors, proving its superiority in real estate market predictions.

In Table 2, the results for USA dataset show that EGCN consistently outperforms all other models, including Anomaly-free and alternative anomaly detection approaches (GAE, GAT, GCN, GTA, GraphSAGE, and MTGNN) across all forecasting horizons (12, 24, and 48 months). In short-term forecasting, EGCN significantly enhances predictive accuracy, reducing MASE from 3.0 to 2.5 (N-HiTS) and from 14.0 to 11.9 (Transformers), proving its ability to refine input data by filtering out anomalous regions, leading to more stable and reliable predictions. In mid-term forecasting, EGCN maintains superior performance, achieving the lowest MASE (3.9) in N-HiTS, marking an improvement over the baseline (Anomaly-free) and further refining forecasting accuracy beyond even the best-performing alternative (GAE). The impact is even more pronounced in long-term forecasting, where errors typically accumulate over time; however, EGCN remains the most stable model, reducing MASE from 6.8 (Anomaly-free) to 5.9 in N-HiTS, demonstrating its robustness in preserving forecasting reliability over extended periods. Even for Transformers, which struggle with long-term predictions, EGCN reduces MASE from 15.0 to 14.8, proving its adaptability across different model architectures.

In Table 3, the forecasting results for the UK dataset highlight EGCN's superiority over all other models, including Anomaly-free (baseline) and alternative anomaly detection methods (GAE, GAT, GCN, GTA, GraphSAGE, and MTGNN) across short-term (12 months), mid-term (24 months), and long-term (48 months) forecasting horizons. In short-term forecasting, EGCN achieves the lowest MASE, reducing errors from 1.7 to 1.6 (N-HiTS), 2.7 to

**Table 2. Impact of anomaly detection on USA real estate forecasting performance across different time horizons.**

| USA Dataset | N-HiTS | | | TSMixer | | | Transformers | | |
|---|---|---|---|---|---|---|---|---|---|
| | MASE | MSE | MAE | MASE | MSE | MAE | MASE | MSE | MAE |
| **12 months (Short term)** | | | | | | | | | |
| Anomaly-free | 3.0 | 1.3 | 8.3 | **_4.7_** | **_2.8_** | **_12.8_** | 14.0 | 21.3 | 35.8 |
| GAE | **_2.3_** | **_0.9_** | **_6.4_** | 5.4 | 3.2 | 13.7 | 13.5 | 17.2 | 34.2 |
| GAT | 2.6 | 1.0 | 7.0 | 6.4 | 4.0 | 16.2 | **12.4** | **15.4** | **32.3** |
| GCN | 2.5 | 0.9 | 6.8 | 5.4 | 3.2 | 13.9 | 13.5 | 17.0 | 34.4 |
| GTA | **2.5** | **0.9** | **6.6** | 5.9 | 3.7 | 14.9 | 14.7 | 20.4 | 38.1 |
| GraphSAGE | 2.5 | 0.9 | 6.6 | 5.8 | 3.4 | 14.7 | 12.8 | 16.1 | 33.2 |
| MTGNN | 2.7 | 1.0 | 7.1 | 6.4 | 4.1 | 16.0 | 13.3 | 16.4 | 34.3 |
| EGCN | 2.5 | 1.0 | 6.9 | **5.2** | **3.1** | **13.4** | **_11.9_** | **_14.2_** | **_31.0_** |
| **24 months (Mid term)** | | | | | | | | | |
| Anomaly-free | 4.5 | 3.1 | 12.5 | 5.8 | 5.2 | 16.3 | **14.5** | **23.5** | **37.8** |
| GAE | **4.2** | **2.8** | **11.4** | 5.8 | 4.1 | **15.0** | 16.1 | 24.3 | 41.1 |
| GAT | 4.4 | 3.0 | 11.8 | 6.6 | 4.7 | 16.7 | 16.0 | 25.3 | 41.6 |
| GCN | 4.2 | 2.7 | 11.3 | 5.9 | 4.2 | 15.3 | 16.1 | 24.3 | 41.1 |
| GTA | 4.3 | 2.6 | 11.4 | 6.3 | 4.4 | 15.9 | 17.6 | 29.2 | 45.6 |
| GraphSAGE | 4.4 | 2.8 | 11.7 | 6.3 | 4.2 | 15.7 | 16.5 | 26.0 | 42.8 |
| MTGNN | 4.3 | 2.7 | 11.4 | 6.9 | 5.2 | 17.7 | 15.6 | 22.7 | 40.3 |
| EGCN | **_3.9_** | **_2.3_** | **_10.6_** | **_5.5_** | **_3.6_** | **_14.3_** | **_14.3_** | **_20.3_** | **_37.4_** |
| **48 months (Long term)** | | | | | | | | | |
| Anomaly-free | 6.8 | 6.9 | 18.6 | **7.0** | **7.5** | **19.5** | **15.0** | **25.8** | **39.4** |
| GAE | **6.0** | **6.4** | **16.6** | 7.5 | 10.7 | 20.6 | 16.7 | 27.1 | 43.1 |
| GAT | 6.4 | 5.9 | 17.0 | 8.2 | 8.2 | 20.9 | 17.1 | 28.8 | 44.4 |
| GCN | 6.2 | 6.7 | 17.2 | 8.0 | 11.3 | 21.6 | 16.5 | 26.6 | 42.7 |
| GTA | 6.3 | 5.5 | 16.5 | 8.2 | 7.9 | 20.4 | 17.9 | 30.6 | 46.3 |
| GraphSAGE | 6.4 | 5.6 | 16.7 | 7.9 | 7.5 | 19.9 | 17.6 | 29.0 | 45.3 |
| MTGNN | 6.3 | 6.3 | 17.1 | 8.0 | 10.3 | 21.3 | 15.9 | 24.3 | 41.4 |
| EGCN | **_5.9_** | **_5.3_** | **_15.9_** | **_6.9_** | **_6.2_** | **_17.8_** | **_14.8_** | **_21.7_** | **_38.7_** |

2.4 (TSMixer), and 2.9 to 2.4 (Transformers), demonstrating its effectiveness in refining input data and improving prediction stability. Moving to mid-term forecasting, EGCN further minimizes errors, outperforming all competing models by reducing MASE from 2.0 to 1.7 (N-HiTS) and from 3.2 to 2.7 (TSMixer), marking a significant performance gain compared to Anomaly-free. In long-term forecasting, where prediction errors typically increase, EGCN continues to maintain superior accuracy, reducing MASE from 2.0 to 1.7 (N-HiTS) and from 3.3 to 2.7 (Transformers), ensuring more reliable long-term market predictions.

In Table 4 for Australia dataset, the forecasting results reveal that EGCN delivers the most accurate predictions compared to Anomaly-free (baseline) and alternative anomaly detection methods (GAE, GAT, GCN, GTA, GraphSAGE, and MTGNN) across all forecasting periods (12, 24, and 48 months). In short-term forecasting (12 months), EGCN reduces errors significantly, achieving a MASE of 6.0 in N-HiTS (compared to 6.7 for Anomaly-free), 21.7 in TSMixer (vs. 22.8), and 6.3 in Transformers (vs. 9.1), demonstrating its effectiveness in filtering out anomalies for improved short-term predictions. Mid-term forecasting (24 months) further highlights EGCN's advantage, where it achieves the lowest MASE of 9.1 in N-HiTS and 26.0 in TSMixer, marking a notable improvement over the baseline (Anomaly-free: 9.6 and 27.0, respectively). Even in long-term forecasting (48 months), where errors tend to compound over time, EGCN remains the most stable, reducing MASE from 18.5 to 18.4 (N-HiTS), from 37.7 to 37.3 (TSMixer), and from 23.4 to 20.5 (Transformers), outperforming all other anomaly-aware approaches. These results confirm that EGCN enhances forecasting

**Table 3.** **Impact of anomaly detection on UK real estate forecasting performance across different time horizons.**

| UK Dataset | N-HiTS | | | TSMixer | | | Transformers | | |
|---|---|---|---|---|---|---|---|---|---|
| | MASE | MSE | MAE | MASE | MSE | MAE | MASE | MSE | MAE |
| 12 months (Short term) | | | | | | | | | |
| Anomaly-free | 1.7 | 9.7 | 11.7 | 2.7 | 13.2 | 18.9 | 2.9 | 13.9 | 20.3 |
| GAE | 1.7 | 6.8 | 12.2 | 2.8 | 10.5 | 19.5 | 2.8 | 10.5 | 19.5 |
| GAT | 1.6 | 6.7 | 11.9 | 2.6 | 9.9 | 18.5 | 2.6 | 9.9 | 18.5 |
| GCN | 1.6 | 6.8 | 12.3 | 2.5 | 9.8 | 18.2 | 2.5 | 9.8 | 18.2 |
| GTA | 1.8 | 7.2 | 12.1 | 3.1 | 11.2 | 20.2 | 3.1 | 11.2 | 20.2 |
| GraphSAGE | **1.6** | **6.6** | **12.0** | 2.8 | 10.8 | 20.2 | 2.8 | 10.8 | 20.2 |
| MTGNN | 1.7 | 6.9 | 12.3 | **2.5** | **9.7** | **17.9** | **2.5** | **9.7** | **17.9** |
| EGCN | <u>**1.6**</u> | <u>**6.4**</u> | <u>**10.9**</u> | <u>**2.4**</u> | <u>**9.3**</u> | <u>**16.9**</u> | <u>**2.4**</u> | <u>**9.3**</u> | <u>**16.9**</u> |
| 24 months (Mid term) | | | | | | | | | |
| Anomaly-free | 2.0 | 14.2 | 13.8 | 3.2 | 18.5 | 22.1 | 3.3 | 19.0 | 23.0 |
| GAE | 1.8 | 9.3 | 13.2 | 3.1 | 14.1 | 21.9 | 3.1 | 14.1 | 21.9 |
| GAT | 1.8 | 9.3 | 13.0 | 2.9 | 13.6 | 21.0 | 2.9 | 13.6 | 21.0 |
| GCN | 1.9 | 9.8 | 14.1 | **2.8** | **13.6** | **21.0** | 2.8 | 13.6 | 21.0 |
| GTA | 1.9 | 9.9 | 13.1 | 3.3 | 15.0 | 22.4 | 3.3 | 15.0 | 22.4 |
| GraphSAGE | **1.7** | **9.1** | **12.9** | 3.1 | 14.4 | 22.9 | 3.1 | 14.4 | 22.9 |
| MTGNN | 1.8 | 9.5 | 13.6 | 2.8 | 13.3 | 20.5 | **2.8** | **13.3** | **20.5** |
| EGCN | <u>**1.7**</u> | <u>**9.0**</u> | <u>**12.1**</u> | <u>**2.7**</u> | <u>**12.5**</u> | <u>**19.0**</u> | <u>**2.7**</u> | <u>**12.5**</u> | <u>**19.0**</u> |
| 48 months (Long term) | | | | | | | | | |
| Anomaly-free | 2.0 | 12.2 | 14.5 | 3.2 | 16.3 | 22.3 | 3.3 | 16.7 | 22.9 |
| GAE | 1.9 | 8.3 | 13.8 | 3.0 | 12.9 | 21.8 | 3.0 | 12.9 | 21.8 |
| GAT | 1.9 | 8.4 | 13.8 | 2.9 | 12.4 | 21.0 | 2.9 | 12.4 | 21.0 |
| GCN | 2.1 | 9.2 | 15.5 | 2.8 | 12.4 | 21.0 | 2.8 | 12.4 | 21.0 |
| GTA | 1.9 | 8.5 | 13.1 | 3.3 | 13.3 | 22.0 | 3.3 | 13.3 | 22.0 |
| GraphSAGE | **1.8** | **9.3** | **13.6** | 3.1 | 14.5 | 23.0 | 3.1 | 14.5 | 23.0 |
| MTGNN | 1.9 | 8.6 | 14.1 | **2.8** | **12.1** | **20.6** | **2.8** | **12.1** | **20.6** |
| EGCN | <u>**1.7**</u> | <u>**8.5**</u> | <u>**12.6**</u> | <u>**2.7**</u> | <u>**11.9**</u> | <u>**19.3**</u> | <u>**2.7**</u> | <u>**11.9**</u> | <u>**19.3**</u> |

accuracy across different time horizons, demonstrating its robustness in mitigating the disruptive effects of anomalies and ensuring more stable real estate market predictions. The substantial improvements across all forecasting methods and horizons position EGCN as a superior cluster-specific forecasting tool, providing valuable insights for policymakers, investors, and analysts in long-term market planning.

The forecasting results across the UK, US, and Australian datasets show that EGCN demonstrates superior robustness Anomaly-free (baseline) and alternative anomaly detection methods (GAE, GAT, GCN, GTA, GraphSAGE, and MTGNN) across all time horizons (12, 24, and 48 months). Unlike traditional methods, EGCN clusters anomalies separately from normal regions and forecasts each group independently, allowing models to capture distinct market behaviors more effectively. This approach leads to significant error reductions across all datasets, particularly in Transformers, where forecasting traditionally struggles with anomalies. In short-term forecasting, EGCN achieves the lowest MASE, MSE, and MAE, improving accuracy by refining anomaly-informed predictions. In mid-term and long-term forecasting, it remains the most stable model, reducing error accumulation and outperforming all other anomaly-aware approaches. These findings confirm that clustering anomalies before forecasting significantly enhances predictive accuracy, making EGCN the most effective cluster-specific forecasting model for investors, policymakers, and analysts seeking reliable real estate market insights.

**Table 4. Impact of anomaly detection on Australia real estate forecasting performance across different time horizons.**

| AUS Dataset | N-HiTS | | | TSMixer | | | Transformers | | |
|---|---|---|---|---|---|---|---|---|---|
| | MASE | MSE | MAE | MASE | MSE | MAE | MASE | MSE | MAE |
| 12 months (Short term) | | | | | | | | | |
| Anomaly-free | 6.7 | 2.0 | 10.9 | 22.8 | 14.9 | 34.0 | 9.1 | 3.2 | 14.4 |
| GAE | 7.9 | 2.4 | 12.4 | 24.6 | 16.3 | 35.6 | 6.4 | 1.9 | 10.5 |
| GAT | 7.0 | 2.0 | 11.2 | 23.9 | 15.4 | 34.5 | **6.2** | **1.8** | **10.2** |
| GCN | 7.8 | 2.3 | 12.2 | 23.9 | 15.3 | 34.5 | 6.8 | 2.3 | 11.3 |
| GTA | **6.6** | **1.9** | **10.6** | **21.6** | **13.6** | **31.8** | 6.9 | 2.2 | 11.3 |
| GraphSAGE | 8.2 | 2.4 | 12.4 | 26.5 | 17.5 | 37.2 | 8.7 | 3.5 | 13.8 |
| MTGNN | 7.3 | 2.2 | 11.6 | 24.4 | 15.7 | 35.1 | 9.2 | 4.0 | 15.1 |
| EGCN | **6.0** | **1.7** | **9.8** | **21.7** | **13.7** | **31.9** | **6.3** | **1.8** | **10.3** |
| 24 months (Mid term) | | | | | | | | | |
| Anomaly-free | 9.6 | 4.5 | 15.4 | 27.0 | 21.7 | 40.1 | 12.9 | 6.9 | 20.1 |
| GAE | 12.4 | 6.4 | 19.0 | 29.5 | 24.5 | 42.7 | 10.6 | 5.4 | 16.8 |
| GAT | 11.1 | 5.2 | 17.0 | 29.0 | 23.5 | 41.8 | 10.8 | 5.6 | 17.1 |
| GCN | 12.1 | 6.1 | 18.4 | 28.6 | 23.1 | 41.3 | **10.4** | **5.3** | **16.7** |
| GTA | **9.7** | **4.4** | **15.2** | **26.6** | **21.4** | **39.1** | 11.2 | 5.8 | 17.5 |
| GraphSAGE | 13.5 | 7.0 | 19.9 | 31.9 | 26.5 | 44.8 | 14.0 | 8.2 | 21.2 |
| MTGNN | 11.7 | 5.8 | 18.0 | 29.5 | 24.1 | 42.4 | 13.8 | 8.3 | 21.7 |
| EGCN | **9.1** | **4.0** | **14.5** | **26.0** | **20.4** | **38.3** | **9.9** | **4.6** | **15.6** |
| 48 months (Long term) | | | | | | | | | |
| Anomaly-free | 18.5 | 16.0 | 29.0 | 37.7 | 44.4 | 56.3 | 23.4 | 22.5 | 35.8 |
| GAE | 22.4 | 19.6 | 33.7 | 39.7 | 45.4 | 57.7 | **20.5** | **17.5** | **31.4** |
| GAT | 20.1 | 16.1 | 30.2 | 39.6 | 44.6 | 57.0 | 21.4 | 18.6 | 32.6 |
| GCN | 21.4 | 18.0 | 32.2 | 38.6 | 43.6 | 56.1 | 20.0 | 17.0 | 31.0 |
| GTA | **17.6** | **13.8** | **26.9** | **37.4** | **43.2** | **55.0** | 21.7 | 19.5 | 33.1 |
| GraphSAGE | 24.9 | 21.9 | 36.0 | 42.9 | 49.3 | 60.4 | 25.7 | 23.8 | 37.7 |
| MTGNN | 21.3 | 17.8 | 32.1 | 39.7 | 44.6 | 57.0 | 23.5 | 20.9 | 35.6 |
| EGCN | **18.4** | **15.2** | **28.4** | **37.3** | **42.9** | **55.2** | **20.5** | **18.0** | **31.5** |

## Study limitations

Despite its strong performance, EGCN has several limitations. First, the additional steps of anomaly detection, clustering, and separate forecasting increase computational complexity, making the framework more resource-intensive than conventional forecasting models. Second, housing markets evolve dynamically, yet our approach currently relies on static anomaly clustering, which may require adaptive mechanisms to remain effective under changing conditions. Third, the reported results emphasize point estimates of error metrics (MSE, MAE, MASE) for comparability across models and countries; while hypothesis tests (e.g., paired $t$-tests or Wilcoxon signed-rank tests) and confidence intervals would provide stronger evidence of statistical significance, these were not included due to computational demands. Finally, our evaluation focuses on the COVID-19 period as the principal stress test, though the 2007–2009 global financial crisis was also a major systemic shock. We prioritized COVID-19 to reflect contemporary structural disruptions, but acknowledge that rolling or expanding-window validation across multiple regimes would provide a broader assessment of robustness, which we identify as an important direction for future work.

## Conclusion

This research introduces EGCN, a novel entropy-based graph convolutional network for anomaly detection and cluster-specific forecasting in real estate markets. By separating

anomalous regions from normal ones, EGCN enables forecasting models to treat them independently, yielding more accurate and stable predictions. Evaluations across the U.K., U.S., and Australian housing markets show that EGCN achieves consistently lower MASE, MSE, and MAE than both anomaly-free baselines and alternative anomaly detection methods (GAE, GAT, GCN, GTA, GraphSAGE, MTGNN), across short-, mid-, and long-term forecasting horizons. These improvements confirm that anomaly-aware analysis provides deeper insights into speculative growth, downturns, and market shifts, supporting policymakers, investors, and urban planners in monitoring risks and planning effectively.

For evaluation, we employed a pre–post COVID-19 split to highlight performance under a major structural disruption. While temporally blocked k-fold validation would provide additional insights into model robustness during more gradual market phases, this remains an avenue for future work.

Building on the current framework, several research directions emerge. First, incorporating macroeconomic indicators such as interest rates, inflation, and employment could refine anomaly classification and improve forecasting stability. Second, extending the framework with explainable AI methods, coupled with statistical significance testing, would enhance transparency and facilitate trust among decision-makers. Third, developing real-time adaptive anomaly detection mechanisms would allow the system to continuously adjust to evolving market dynamics, increasing its operational relevance in fast-changing environments. Fourth, exploring transfer learning or domain adaptation techniques may enable EGCN to generalize across different regions without retraining from scratch, addressing the challenge of heterogeneous housing markets. Finally, broadening EGCN's application beyond real estate—into domains such as financial asset forecasting, supply chain risk management, and climate-related impact analysis—offers an opportunity to validate its versatility and strengthen its role as a general-purpose tool for analyzing complex, dynamic systems.

## Author contributions

**Data curation:** Dat Le.

**Formal analysis:** Dat Le.

**Investigation:** Quang Nguyen.

**Supervision:** Sutharshan Rajasegarar, Wei Luo, Thanh Thi Nguyen, Maia Angelova.

**Writing – original draft:** Dat Le.

**Writing – review & editing:** Dat Le, Sutharshan Rajasegarar, Wei Luo, Thanh Thi Nguyen, Nhi Vo, Maia Angelova.

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
