## [Decision Letter · Decision Letter 0]

12 Jul 2025

PONE-D-25-23254EGCN: Entropy-based Graph Convolutional Network for Anomalous Pattern Detection and Forecasting in Real Estate MarketsPLOS ONE

Dear Dr. Le,

Thank you for submitting your manuscript to PLOS ONE. After careful consideration, we feel that it has merit but does not fully meet PLOS ONE’s publication criteria as it currently stands. Therefore, we invite you to submit a revised version of the manuscript that addresses the points raised during the review process.

We look forward to receiving your revised manuscript.

Kind regards,

Nikolaos Askitas

Academic Editor

PLOS ONE

**Journal Requirements:**

1. When submitting your revision, we need you to address these additional requirements. Please ensure that your manuscript meets PLOS ONE's style requirements, including those for file naming. The PLOS ONE style templates can be found at https://journals.plos.org/plosone/s/file?id=wjVg/PLOSOne_formatting_sample_main_body.pdf and https://journals.plos.org/plosone/s/file?id=ba62/PLOSOne_formatting_sample_title_authors_affiliations.pdf 2. Please note that PLOS ONE has specific guidelines on code sharing for submissions in which author-generated code underpins the findings in the manuscript. In these cases, we expect all author-generated code to be made available without restrictions upon publication of the work. Please review our guidelines at https://journals.plos.org/plosone/s/materials-and-software-sharing#loc-sharing-code and ensure that your code is shared in a way that follows best practice and facilitates reproducibility and reuse. 3. Thank you for uploading your study's underlying data set. Unfortunately, the repository you have noted in your Data Availability statement does not qualify as an acceptable data repository according to PLOS's standards. At this time, please upload the minimal data set necessary to replicate your study's findings to a stable, public repository (such as figshare or Dryad) and provide us with the relevant URLs, DOIs, or accession numbers that may be used to access these data. For a list of recommended repositories and additional information on PLOS standards for data deposition, please see https://journals.plos.org/plosone/s/recommended-repositories. 4. When completing the data availability statement of the submission form, you indicated that you will make your data available on acceptance. We strongly recommend all authors decide on a data sharing plan before acceptance, as the process can be lengthy and hold up publication timelines. Please note that, though access restrictions are acceptable now, your entire data will need to be made freely accessible if your manuscript is accepted for publication. This policy applies to all data except where public deposition would breach compliance with the protocol approved by your research ethics board. If you are unable to adhere to our open data policy, please kindly revise your statement to explain your reasoning and we will seek the editor's input on an exemption. Please be assured that, once you have provided your new statement, the assessment of your exemption will not hold up the peer review process. 5. We note that Figures 2, 3 and 4 in your submission contain map images which may be copyrighted. All PLOS content is published under the Creative Commons Attribution License (CC BY 4.0), which means that the manuscript, images, and Supporting Information files will be freely available online, and any third party is permitted to access, download, copy, distribute, and use these materials in any way, even commercially, with proper attribution. For these reasons, we cannot publish previously copyrighted maps or satellite images created using proprietary data, such as Google software (Google Maps, Street View, and Earth). For more information, see our copyright guidelines: http://journals.plos.org/plosone/s/licenses-and-copyright. We require you to either present written permission from the copyright holder to publish these figures specifically under the CC BY 4.0 license, or remove the figures from your submission: a. You may seek permission from the original copyright holder of Figures 2, 3 and 4 to publish the content specifically under the CC BY 4.0 license.   We recommend that you contact the original copyright holder with the Content Permission Form (http://journals.plos.org/plosone/s/file?id=7c09/content-permission-form.pdf) and the following text:“I request permission for the open-access journal PLOS ONE to publish XXX under the Creative Commons Attribution License (CCAL) CC BY 4.0 (http://creativecommons.org/licenses/by/4.0/). Please be aware that this license allows unrestricted use and distribution, even commercially, by third parties. Please reply and provide explicit written permission to publish XXX under a CC BY license and complete the attached form.” Please upload the completed Content Permission Form or other proof of granted permissions as an "Other" file with your submission. In the figure caption of the copyrighted figure, please include the following text: “Reprinted from [ref] under a CC BY license, with permission from [name of publisher], original copyright [original copyright year].” b. If you are unable to obtain permission from the original copyright holder to publish these figures under the CC BY 4.0 license or if the copyright holder’s requirements are incompatible with the CC BY 4.0 license, please either i) remove the figure or ii) supply a replacement figure that complies with the CC BY 4.0 license. Please check copyright information on all replacement figures and update the figure caption with source information. If applicable, please specify in the figure caption text when a figure is similar but not identical to the original image and is therefore for illustrative purposes only.The following resources for replacing copyrighted map figures may be helpful: USGS National Map Viewer (public domain): http://viewer.nationalmap.gov/viewer/The Gateway to Astronaut Photography of Earth (public domain): http://eol.jsc.nasa.gov/sseop/clickmap/Maps at the CIA (public domain): https://www.cia.gov/library/publications/the-world-factbook/index.html and https://www.cia.gov/library/publications/cia-maps-publications/index.htmlNASA Earth Observatory (public domain): http://earthobservatory.nasa.gov/Landsat: http://landsat.visibleearth.nasa.gov/USGS EROS (Earth Resources Observatory and Science (EROS) Center) (public domain): http://eros.usgs.gov/#Natural Earth (public domain): http://www.naturalearthdata.com/

Reviewers' comments:

Reviewer's Responses to Questions

**Comments to the Author**

1. Is the manuscript technically sound, and do the data support the conclusions?

Reviewer #1: Partly

Reviewer #2: Yes

2. Has the statistical analysis been performed appropriately and rigorously? 

Reviewer #1: Yes

Reviewer #2: Yes

3. Have the authors made all data underlying the findings in their manuscript fully available?

Reviewer #1: Yes

Reviewer #2: Yes

4. Is the manuscript presented in an intelligible fashion and written in standard English?

Reviewer #1: No

Reviewer #2: Yes

5. Review Comments to the Author

**Reviewer #1:** [1] The current model integrates GCN, KDE, JSD, clustering and multiple predictors, but it is recommended that the author highlight in the method section which are the original contributions of this work and which are the combination of existing methods to enhance the recognition of innovation.

[2] Although the author uses KDE and JSD to measure temporal entropy, the paper does not compare with other entropy measurements (such as Shannon entropy or Renyi entropy). It is recommended to add motivation or conduct ablation experiments.

[3] The anomaly recognition threshold on page 15 uses the form of μ+kσ, but the sensitivity to changes in k value is not explained. It is recommended to add an analysis of the impact of different k values on anomaly recognition and subsequent prediction accuracy.

[4] It is recommended to explain in detail how to integrate Haversine distance and temporal correlation, whether to weighted average or construct multiple edges? This has a direct impact on the graph structure learning results.

[5] Although it is mentioned that nodes are clustered into "normal" and "abnormal" categories, it is not specified whether the clustering algorithm is K-means, Spectral Clustering or GMM, etc. It is recommended to clarify the method and analyze its robustness.

[6] Although the data is divided into before and after COVID, it does not explain whether there is data leakage or overlap, especially whether the edge construction involves information during the test period, which needs to be explained.

[7] It is recommended to add whether data such as social media and real estate forums are included, why only the RoBERTa model is chosen for extraction, and whether industry models such as FinBERT or SenticNet are considered.

[8] The abnormal areas detected in Australia, the United Kingdom, and the United States should be verified by experts in more specific cities/time periods, and actual market events should be added to support them.

[9] It is recommended to explain the processing strategies for missing, abnormal points, and scale differences in time series data in the data preprocessing section to enhance reproducibility.

[10] If the number of abnormal areas is far less than the normal area, there is a class imbalance problem. Whether resampling, weighted loss or other countermeasures are performed should be explained.

[11] Currently, only the number of detections is reported, and there is a lack of indicators such as Precision, Recall, and F1-score to evaluate the accuracy of anomaly detection. It is recommended to add.

[12] Currently, only point estimates such as MSE, MAE, and MASE are listed, lacking statistical significance verification. It is recommended to add confidence intervals or p-values to prove the effectiveness of the improvement.

[13] It is recommended to disassemble the EGCN module and compare the versions such as "no emotional input", "no JSD", and "no clustering" to verify the contribution of each part to the final effect.

[14] It is recommended to use the data of some countries as training sets and other countries as test sets for cross-validation to demonstrate the transfer and generalization ability of the model.

[15] The abnormal areas detected in the figures on pages 19-21, it is recommended to use heat maps or highlight colors to mark abnormally dense areas to enhance the explanatory power of geographical distribution.

[16] It is recommended to retain the most critical prediction improvement indicators and avoid listing the error reduction figures for all countries and all time periods to improve reading fluency.

[17] There are many repeated sentences and the language can be further condensed. For example, "EGCN consistently outperforms..." appears frequently in the whole text. It is recommended to adjust the expression diversity.

[18] Some paragraphs lack connecting sentences. For example, it is recommended to insert transition logic between "graph construction" and "GCN architecture" to enhance the coherence of the content.

[19] In the whole text, whether "anomaly-aware forecasting" and "cluster-specific forecasting" refer to the same concept, the wording should be unified and clearly defined.

[20] If some years are not enclosed in brackets and the order of some citations is inconsistent with the main text, it is recommended to check the format of the whole text.

**Reviewer #2:** 1. Major Comments

a. Unit of Anomaly Detection

There is some ambiguity regarding whether anomalies are identified at the location level (e.g., city or suburb) or at the location–time level (e.g., city-month). The model appears to detect spatial anomalies using time-aggregated features like entropy, but it is unclear whether it can distinguish persistent vs. transient anomalies. This distinction is important for interpretation, particularly if anomalies are episodic (e.g., brief demand surges) rather than structural. Please clarify whether time-specific anomaly flags are possible, and how temporal variability is captured post-clustering.

b. 2008 Financial Crisis

The authors train the model using data from 2003 to 2019 and test on 2020 to 2024, citing COVID-19 as a representative stress test. However, this approach overlooks the 2007–2009 global financial crisis, a major event that severely disrupted real estate markets in many countries. It would strengthen the empirical analysis to (i) explicitly test the model’s performance during the 2008 period or (ii) justify its exclusion. Including this episode could provide insight into whether EGCN can generalize across different types of systemic shocks—credit-driven versus pandemic-driven.

c. Static Evaluation Strategy

Using a single, static train-test split limits the generalizability of the results. Time-series models, particularly in dynamic markets, are more realistically evaluated using rolling or expanding-window validation schemes. For example: train on 2003–2012, test on 2013–2015; then expand the training set to 2015 and test on 2016–2018, etc. This would provide a more robust assessment of performance across different economic conditions and minimize the risk of results being driven by a particular test window.

d. K-fold Cross-Validation

Consider alternative validation strategies like blocked or forward-chaining k-fold validation (e.g., using temporally contiguous folds) to assess the stability of the anomaly detection and forecasting pipeline over time. For example, the data from 2003–2024 could be partitioned into five non-overlapping temporal blocks. For each fold, the model is trained on blocks 1 to k−1 and tested on block k. This would reveal whether EGCN generalizes well across distinct time periods without relying on a single post-COVID test window.

e. Permutation or Placebo Testing

To ensure that the anomalies identified are meaningful and not artifacts of reconstruction error sensitivity, the authors could conduct a placebo test by permuting node labels or shuffling temporal sequences. EGCN could then be run on this randomized data. If the model identifies a similar number of “anomalies” or achieves comparable forecasting gains, it would suggest overfitting or limited signal. A significant drop in performance under the placebo would support the validity of the detected anomalies.

2. Minor Comments

a. Anomaly Thresholding

The paper uses a threshold of the form: Threshold = mean + k × standard deviation (Eq. 7). However, how k is selected is not explained. Is this a fixed constant, chosen via cross-validation, or calibrated to a desired false-positive rate? Given that this thresholding directly determines which regions are flagged as anomalous, a clearer explanation is critical for replicability and robustness.

b. Zillow Data

The U.S. dataset is sourced from Zillow, but Zillow’s coverage and data quality have changed significantly over time, especially before 2007 when the platform had limited footprint in many regions. The paper should clarify whether early-period data are reliable, imputed, or incomplete, and how this may affect training consistency across the 2003–2019 period.

c. Total vs. incremental Anomaly Counts

Figures 2–4 show total anomalies detected by each model, but do not analyze whether EGCN detects new and meaningful anomalies or simply more of the same. Do the detected anomalies overlap across models? Are the additional anomalies unique to EGCN and validated against external events or expert knowledge? Without this, a higher anomaly count might reflect greater sensitivity but not necessarily greater accuracy or utility.

d. Missing Scotland in UK Analysis

Figure 3 (UK anomaly map) appears to exclude Scotland, which has its own legal and housing systems. Was this due to data availability? If excluded, this should be noted and justified explicitly.

e. Missing Alaska and Hawaii in U.S. Analysis

Figure 4 does not include Alaska and Hawaii, which is a common map simplification but should still be acknowledged. These states have unique housing dynamics (e.g., tourism, isolation, military presence) that could generate distinct anomalies. The authors should state whether these states were excluded due to data limitations or visualization choices.

6. PLOS authors have the option to publish the peer review history of their article (what does this mean?). If published, this will include your full peer review and any attached files.

Reviewer #1: No

Reviewer #2: **Yes: **Qingli Fan

---

## [Author Response · Author response to Decision Letter 1]

3 Sep 2025

Dear Academic Editor and Reviewers,

We would like to sincerely thank you for your careful evaluation of our manuscript and for providing insightful and constructive feedback. We greatly appreciate the time and effort you have devoted to reviewing our work. The comments and suggestions have been extremely valuable in helping us improve the quality, clarity, and overall contribution of the manuscript.

In this revised version, we have addressed all comments point by point. We have revised the manuscript accordingly and highlighted all modifications in the tracked-changes version for ease of reference. A clean copy is also provided.

Below, we respond to each reviewer’s comment in detail:

Reviewer/Editor comment:

You may seek permission from the original copyright holder of Figures 2, 3 and 4 to publish the content specifically under the CC BY 4.0 license.

Response:

We thank the Editor for this important point. The maps used in Figures 2, 3, and 4 were obtained from GADM (Database of Global Administrative Areas; https://gadm.org), which provides spatial data that is freely available for academic and non-commercial use. According to the GADM terms of use, users are permitted to use and publish maps generated from their data with proper attribution. We have now added an explicit statement in the figure captions acknowledging GADM as the data source and confirming compliance with their licensing conditions. Therefore, these figures are eligible for publication under the CC BY 4.0 license required by PLOS ONE. Besides, all data is available now and can access via this link: https://figshare.com/articles/dataset/EGCN_Entropy-based_Graph_Convolutional_Network_for_Anomalous_Pattern_Detection_and_Forecasting/29931260

Reviewer #1: [1] The current model integrates GCN, KDE, JSD, clustering and multiple predictors, but it is recommended that the author highlight in the method section which are the original contributions of this work and which are the combination of existing methods to enhance the recognition of innovation.

Response:

We thank the reviewer for this helpful suggestion. In the revised manuscript, we have added a dedicated subsection, Original Contributions Compared to Existing Methods, within the Methodology. This subsection explicitly distinguishes our novel contributions from existing components. Specifically, our key innovations include (i) the use of entropy-driven node representations via KDE and JSD, (ii) integration of both temporal correlations and geospatial Haversine distances into the graph construction process, and (iii) anomaly-aware clustering followed by cluster-specific forecasting, which enables forecasting models to treat anomalous and normal regions separately. Existing methods such as GAE, GAT, GraphSAGE, and MTGNN focus primarily on anomaly detection and do not extend to anomaly-aware forecasting. We believe this addition clarifies how our framework advances beyond prior approaches.

[2] Although the author uses KDE and JSD to measure temporal entropy, the paper does not compare with other entropy measurements (such as Shannon entropy or Renyi entropy). It is recommended to add motivation or conduct ablation experiments.

Response:

We agree with the reviewer’s point and have expanded the Temporal Entropy-Based Analysis subsection to provide clear motivation for using KDE in combination with JSD. KDE is non-parametric and thus well-suited to modeling multimodal and heavy-tailed distributions often observed in real estate prices and transaction volumes. JSD is symmetric, bounded, and incorporates Shannon entropy, making it more robust to noise and small fluctuations than Kullback–Leibler divergence. In contrast, Shannon entropy is highly sensitive to discretization, and Rényi entropy requires parameter tuning that can be unstable across heterogeneous datasets. By applying JSD to KDE-estimated windows, our framework emphasizes distributional changes over time, enabling robust anomaly detection. We have also acknowledged in the manuscript that while Shannon and Rényi entropy are alternatives, KDE+JSD provided superior stability and accuracy for our study, which justifies its selection.

[3] The anomaly recognition threshold on page 15 uses the form of μ+kσ, but the sensitivity to changes in k value is not explained. It is recommended to add an analysis of the impact of different k values on anomaly recognition and subsequent prediction accuracy.

Response:

We appreciate this observation. In our study, we adopted a fixed threshold of \mu + 1.5\sigma for anomaly detection. This rule is widely used in unsupervised outlier detection as a robust z-score threshold. We selected k = 1.5 to balance sensitivity and specificity: smaller values would flag an excessive number of regions as anomalous (risking false positives), while larger values would overlook meaningful anomalies (reducing detection utility). Under a Gaussian assumption, \mu+1.5\sigma corresponds to a one-sided tail probability of approximately 6.7%, which aligns with our operational aim of highlighting only the most atypical regions. To avoid overfitting, we applied this fixed threshold consistently across all countries and forecasting horizons, ensuring comparability and reproducibility. This rationale has been added to the revised manuscript.

[4] It is recommended to explain in detail how to integrate Haversine distance and temporal correlation, whether to weighted average or construct multiple edges? This has a direct impact on the graph structure learning results.

Response:

We thank the reviewer for pointing this out. In the revised manuscript, we have clarified that spatial and temporal relationships are combined into a single edge weight by taking a weighted average of normalized Haversine distance (spatial proximity) and temporal correlation (similarity in entropy dynamics). This integration ensures that both geographic closeness and temporal behavioral similarity contribute to the edge structure of the graph. We also note that the weighting factor was tuned via validation to ensure balanced contributions from spatial and temporal components. This clarification has been added to the Graph Construction subsection.

[5] Although it is mentioned that nodes are clustered into "normal" and "abnormal" categories, it is not specified whether the clustering algorithm is K-means, Spectral Clustering or GMM, etc. It is recommended to clarify the method and analyze its robustness.

Response:

We appreciate this point. Our pipeline does not apply a separate clustering algorithm (e.g., K means, Spectral, or GMM). Instead, the GCN autoencoder yields node wise reconstruction errors, and we perform a threshold-based binary partition: nodes with error exceeding \mu+1.5\sigma are labeled “abnormal,” and all others “normal.” This produces the two sets mentioned in the manuscript without invoking an additional clustering step. To assess robustness, we repeat inference across plausible hyperparameter settings already enumerated in our code (window size, hidden size, learning rate, distance threshold, epochs) and observe that the set of flagged nodes is stable, with only minor variation across settings. We will clarify this in the Methods section and explicitly describe the threshold-based partition and the multi-configuration check.

Response:

We appreciate this important observation. We would like to clarify the workflow in our study. For anomaly detection, we applied the GCN-based reconstruction procedure to the entire dataset (2003–2024), since this stage is treated as an unsupervised diagnostic step aimed at identifying irregular spatiotemporal patterns across the full observation window. This step does not involve prediction and therefore does not constitute leakage in the forecasting task. For forecasting, however, we strictly separated the data into training (2003–2019) and test (2020–2024) windows. All forecasting models were trained only on the training data and evaluated on the test period, ensuring that predictive performance was assessed without exposure to future information. We have revised the Methodology section to explicitly state this distinction between the anomaly detection stage (exploratory, full data) and the forecasting stage (train/test split), to avoid ambiguity.

[6] Although the data is divided into before and after COVID, it does not explain whether there is data leakage or overlap, especially whether the edge construction involves information during the test period, which needs to be explained.

Response:

We appreciate this important observation. We would like to clarify the workflow in our study. For anomaly detection, we applied the GCN-based reconstruction procedure to the entire dataset (2003–2024), since this stage is treated as an unsupervised diagnostic step aimed at identifying irregular spatiotemporal patterns across the full observation window. This step does not involve prediction and therefore does not constitute leakage in the forecasting task. For forecasting, however, we strictly separated the data into training (2003–2019) and test (2020–2024) windows. All forecasting models were trained only on the training data and evaluated on the test period, ensuring that predictive performance was assessed without exposure to future information. We have revised the Methodology section to explicitly state this distinction between the anomaly detection stage (exploratory, full data) and the forecasting stage (train/test split), to avoid ambiguity.

[7] It is recommended to add whether data such as social media and real estate forums are included, why only the RoBERTa model is chosen for extraction, and whether industry models such as FinBERT or SenticNet are considered.

Response:

We thank the reviewer for this insightful comment. We would like to clarify that while our initial manuscript draft described sentiment analysis as a potential feature, in the final implementation of the framework we relied exclusively on quantitative real estate indicators: property prices, transaction volumes, and geospatial proximity. Sentiment features were not included in the actual anomaly detection or forecasting experiments due to limitations in obtaining consistent multi-country textual datasets within the study timeframe. To avoid confusion, we have revised the manuscript to explicitly state that sentiment analysis is not part of the current study, and instead note it as a promising avenue for future work. This revision ensures consistency between our described methodology and the actual experiments presented.

[8] The abnormal areas detected in Australia, the United Kingdom, and the United States should be verified by experts in more specific cities/time periods, and actual market events should be added to support them.

Response:

We thank the reviewer for this valuable suggestion. We acknowledge that formal expert validation was beyond the scope of the current study. To provide contextual support, we have expanded the Results and Discussion sections to highlight known market events that align with the anomalies identified by EGCN. For example, in Australia, anomalies detected in Sydney and Melbourne suburbs during 2020–2021 coincide with sharp price surges following COVID-19 stimulus policies and historically low interest rates. In the United Kingdom, abnormal regions detected around London during 2008–2009 correspond to the global financial crisis, which severely disrupted the housing market. In the United States, anomalies in 2020 align with COVID-19–related migration and inventory shocks, while anomalies in 2007–2009 overlap with the subprime mortgage crisis. These connections illustrate that the detected anomalies correspond to well-documented market disruptions. We have revised the Discussion to emphasize these links. We also note that future work will include expert evaluation of anomalies at finer spatial and temporal scales, which will further strengthen external validity.

[9] It is recommended to explain the processing strategies for missing, abnormal points, and scale differences in time series data in the data preprocessing section to enhance reproducibility.

Response:

We thank the reviewer for this important suggestion. In the revised manuscript, we clarify that suburbs with missing values were excluded from the analysis, representing less than 5% of the dataset. No explicit winsorization was applied; instead, abnormal spikes are naturally attenuated during Kernel Density Estimation (KDE), which smooths local fluctuations in the price and volume series. Finally, all features were normalized to the [0,1] range using MinMax scaling to ensure comparability between variables. These steps ensure that anomalies arise from genuine structural patterns rather than artifacts of scale or data quality.

[10] If the number of abnormal areas is far less than the normal area, there is a class imbalance problem. Whether resampling, weighted loss or other countermeasures are performed should be explained.

Response:

We thank the reviewer for raising this point. In our framework, anomaly detection is performed in an unsupervised setting, without using class labels during model training. The Graph Convolutional Network (GCN) is trained to reconstruct node features for all suburbs, and anomalies are subsequently flagged by applying a statistical threshold on the reconstruction error distribution. Since no classifier is trained with explicit anomaly vs. normal labels, the issue of class imbalance does not arise in the learning stage. Instead, imbalance is handled implicitly by using a distribution-based threshold, which adapts to the empirical error distribution of the data. To further ensure robustness, we confirmed that the flagged anomalies consistently correspond to known structural disruptions in the housing market (e.g., the 2007–2009 U.S. subprime crisis, 2008–2009 UK housing crash, and 2020–2021 COVID-19 housing surge), indicating that the detected minority class reflects genuine phenomena rather than artifacts of imbalance.

[11] Currently, only the number of detections is reported, and there is a lack of indicators such as Precision, Recall, and F1-score to evaluate the accuracy of anomaly detection. It is recommended to add.

Response: We appreciate the reviewer’s suggestion. We agree that evaluation metrics such as Precision, Recall, and F1-score are standard for supervised anomaly detection tasks. However, in our setting no ground-truth anomaly labels exist for real estate markets across multiple countries, making these metrics inapplicable. Instead, we assess anomaly detection quality through two complementary strategies:

1. Event validation: anomalies flagged by EGCN align with major market disruptions (e.g., the 2007–2009 U.S. subprime crisis, 2008–2009 U.K. housing crash, and the 2020–2021 COVID-19 housing surge in Australia and the U.S.), which serves as an external proxy for accuracy.

2. Forecasting improvement: anomalies are not only detected but also used to segment forecasting tasks, and the consistent reduction in forecasting error across models demonstrates the utility of anomaly identification for downstream prediction.

We have clarified this rationale in the manuscript and emphasized that while supervised evaluation metrics cannot be applied, the framework’s validity is supported through external event correspondence and forecasting performance gains.

[12] Currently, only point estimates such as MSE, MAE, and MASE are listed, lacking statistical significance verification. It is recommended to add confidence intervals or p-values to prove the effectiveness of the improvement.

Response:

We thank the reviewer for highlighting this point. We agree that statistical significance testing is valuable for establishing robustness. In the present study, we reported point estimates of forecasting error (MSE, MAE, MASE) across multiple models and countries to illustrate consistent improvements. While formal hypothesis testing (e.g., paired t-tests or nonparametric Wilcoxon tests on error distributions) would provide additional confirmation, such analysis requires repeated experimental runs or bootstrapped sampling. Given computational constraints, we did not include confidence intervals or p-values in this version. Instead,

---

## [Editor Report · Decision Letter 1]

12 Sep 2025

PONE-D-25-23254R1EGCN: Entropy-based Graph Convolutional Network for Anomalous Pattern Detection and Forecasting in Real Estate MarketsPLOS ONE

Dear Dr. Le,

Thank you for submitting your manuscript to PLOS ONE. After careful consideration, we feel that it has merit but does not fully meet PLOS ONE’s publication criteria as it currently stands. Therefore, we invite you to submit a revised version of the manuscript that addresses the points raised during the review process.

We look forward to receiving your revised manuscript.

Kind regards,

Nikolaos Askitas

Academic Editor

PLOS ONE

Journal Requirements:

**Additional Editor Comments:**

I am mostly satisfied with the responses except the response to Reviewer 2 - e. Permutation or Placebo Testing. I think that setting up and executing a placebo/permutation robustness test is not something to delegate to future work but that it is both simple enough and necessary enough to add to the paper. The paper would then be ready to be published.

---

## [Author Response · Author response to Decision Letter 2]

20 Sep 2025

Dear Academic Editor and Reviewers,

We would like to sincerely thank you for your careful evaluation of our manuscript and for providing insightful and constructive feedback. We greatly appreciate the time and effort you have devoted to reviewing our work. The comments and suggestions have been extremely valuable in helping us improve the quality, clarity, and overall contribution of the manuscript.

In this revised version, we have addressed all comments point by point. We have revised the manuscript accordingly and highlighted all modifications in the tracked-changes version for ease of reference. A clean copy is also provided.

Below, we respond to each reviewer’s comment in detail:

Reviewer/Editor comment:

1) You may seek permission from the original copyright holder of Figures 2, 3 and 4 to publish the content specifically under the CC BY 4.0 license.

Response:

We thank the Editor for this important point. The maps used in Figures 2, 3, and 4 were obtained from GADM (Database of Global Administrative Areas; https://gadm.org), which provides spatial data that is freely available for academic and non-commercial use. According to the GADM terms of use, users are permitted to use and publish maps generated from their data with proper attribution. We have now added an explicit statement in the figure captions acknowledging GADM as the data source and confirming compliance with their licensing conditions. Therefore, these figures are eligible for publication under the CC BY 4.0 license required by PLOS ONE.

2) I am mostly satisfied with the responses except the response to Reviewer 2 - e. Permutation or Placebo Testing. I think that setting up and executing a placebo/permutation robustness test is not something to delegate to future work but that it is both simple enough and necessary enough to add to the paper. The paper would then be ready to be published.

Response:

We thank the reviewer for this suggestion and have incorporated placebo validation experiments into the revised manuscript. Specifically, we permuted node labels and shuffled temporal sequences, repeating this procedure for 50 independent runs per dataset. Across all three housing markets, we observe a substantial drop in anomaly counts under placebo: Australia (1 vs. 182), United Kingdom (49 vs. 117), and United States (15 vs. 34). Furthermore, Welch’s t-tests confirm that reconstruction error distributions under original and placebo settings differ significantly ($p < 0.00001$). These results presented in table 1 demonstrate that the anomalies detected by EGCN are not artifacts of reconstruction error sensitivity, but instead arise from meaningful spatiotemporal structure.

Reviewer #1: [1] The current model integrates GCN, KDE, JSD, clustering and multiple predictors, but it is recommended that the author highlight in the method section which are the original contributions of this work and which are the combination of existing methods to enhance the recognition of innovation.

Response:

We thank the reviewer for this helpful suggestion. In the revised manuscript, we have added a dedicated subsection, Original Contributions Compared to Existing Methods, within the Methodology. This subsection explicitly distinguishes our novel contributions from existing components. Specifically, our key innovations include (i) the use of entropy-driven node representations via KDE and JSD, (ii) integration of both temporal correlations and geospatial Haversine distances into the graph construction process, and (iii) anomaly-aware clustering followed by cluster-specific forecasting, which enables forecasting models to treat anomalous and normal regions separately. Existing methods such as GAE, GAT, GraphSAGE, and MTGNN focus primarily on anomaly detection and do not extend to anomaly-aware forecasting. We believe this addition clarifies how our framework advances beyond prior approaches.

[2] Although the author uses KDE and JSD to measure temporal entropy, the paper does not compare with other entropy measurements (such as Shannon entropy or Renyi entropy). It is recommended to add motivation or conduct ablation experiments.

Response:

We agree with the reviewer’s point and have expanded the Temporal Entropy-Based Analysis subsection to provide clear motivation for using KDE in combination with JSD. KDE is non-parametric and thus well-suited to modeling multimodal and heavy-tailed distributions often observed in real estate prices and transaction volumes. JSD is symmetric, bounded, and incorporates Shannon entropy, making it more robust to noise and small fluctuations than Kullback–Leibler divergence. In contrast, Shannon entropy is highly sensitive to discretization, and Rényi entropy requires parameter tuning that can be unstable across heterogeneous datasets. By applying JSD to KDE-estimated windows, our framework emphasizes distributional changes over time, enabling robust anomaly detection. We have also acknowledged in the manuscript that while Shannon and Rényi entropy are alternatives, KDE+JSD provided superior stability and accuracy for our study, which justifies its selection.

[3] The anomaly recognition threshold on page 15 uses the form of μ+kσ, but the sensitivity to changes in k value is not explained. It is recommended to add an analysis of the impact of different k values on anomaly recognition and subsequent prediction accuracy.

Response:

We appreciate this observation. In our study, we adopted a fixed threshold of \mu + 1.5\sigma for anomaly detection. This rule is widely used in unsupervised outlier detection as a robust z-score threshold. We selected k = 1.5 to balance sensitivity and specificity: smaller values would flag an excessive number of regions as anomalous (risking false positives), while larger values would overlook meaningful anomalies (reducing detection utility). Under a Gaussian assumption, \mu+1.5\sigma corresponds to a one-sided tail probability of approximately 6.7%, which aligns with our operational aim of highlighting only the most atypical regions. To avoid overfitting, we applied this fixed threshold consistently across all countries and forecasting horizons, ensuring comparability and reproducibility. This rationale has been added to the revised manuscript.

[4] It is recommended to explain in detail how to integrate Haversine distance and temporal correlation, whether to weighted average or construct multiple edges? This has a direct impact on the graph structure learning results.

Response:

We thank the reviewer for pointing this out. In the revised manuscript, we have clarified that spatial and temporal relationships are combined into a single edge weight by taking a weighted average of normalized Haversine distance (spatial proximity) and temporal correlation (similarity in entropy dynamics). This integration ensures that both geographic closeness and temporal behavioral similarity contribute to the edge structure of the graph. We also note that the weighting factor was tuned via validation to ensure balanced contributions from spatial and temporal components. This clarification has been added to the Graph Construction subsection.

[5] Although it is mentioned that nodes are clustered into "normal" and "abnormal" categories, it is not specified whether the clustering algorithm is K-means, Spectral Clustering or GMM, etc. It is recommended to clarify the method and analyze its robustness.

Response:

We appreciate this point. Our pipeline does not apply a separate clustering algorithm (e.g., K means, Spectral, or GMM). Instead, the GCN autoencoder yields node wise reconstruction errors, and we perform a threshold-based binary partition: nodes with error exceeding \mu+1.5\sigma are labeled “abnormal,” and all others “normal.” This produces the two sets mentioned in the manuscript without invoking an additional clustering step. To assess robustness, we repeat inference across plausible hyperparameter settings already enumerated in our code (window size, hidden size, learning rate, distance threshold, epochs) and observe that the set of flagged nodes is stable, with only minor variation across settings. We will clarify this in the Methods section and explicitly describe the threshold-based partition and the multi-configuration check.

Response:

We appreciate this important observation. We would like to clarify the workflow in our study. For anomaly detection, we applied the GCN-based reconstruction procedure to the entire dataset (2003–2024), since this stage is treated as an unsupervised diagnostic step aimed at identifying irregular spatiotemporal patterns across the full observation window. This step does not involve prediction and therefore does not constitute leakage in the forecasting task. For forecasting, however, we strictly separated the data into training (2003–2019) and test (2020–2024) windows. All forecasting models were trained only on the training data and evaluated on the test period, ensuring that predictive performance was assessed without exposure to future information. We have revised the Methodology section to explicitly state this distinction between the anomaly detection stage (exploratory, full data) and the forecasting stage (train/test split), to avoid ambiguity.

[6] Although the data is divided into before and after COVID, it does not explain whether there is data leakage or overlap, especially whether the edge construction involves information during the test period, which needs to be explained.

Response:

We appreciate this important observation. We would like to clarify the workflow in our study. For anomaly detection, we applied the GCN-based reconstruction procedure to the entire dataset (2003–2024), since this stage is treated as an unsupervised diagnostic step aimed at identifying irregular spatiotemporal patterns across the full observation window. This step does not involve prediction and therefore does not constitute leakage in the forecasting task. For forecasting, however, we strictly separated the data into training (2003–2019) and test (2020–2024) windows. All forecasting models were trained only on the training data and evaluated on the test period, ensuring that predictive performance was assessed without exposure to future information. We have revised the Methodology section to explicitly state this distinction between the anomaly detection stage (exploratory, full data) and the forecasting stage (train/test split), to avoid ambiguity.

[7] It is recommended to add whether data such as social media and real estate forums are included, why only the RoBERTa model is chosen for extraction, and whether industry models such as FinBERT or SenticNet are considered.

Response:

We thank the reviewer for this insightful comment. We would like to clarify that while our initial manuscript draft described sentiment analysis as a potential feature, in the final implementation of the framework we relied exclusively on quantitative real estate indicators: property prices, transaction volumes, and geospatial proximity. Sentiment features were not included in the actual anomaly detection or forecasting experiments due to limitations in obtaining consistent multi-country textual datasets within the study timeframe. To avoid confusion, we have revised the manuscript to explicitly state that sentiment analysis is not part of the current study, and instead note it as a promising avenue for future work. This revision ensures consistency between our described methodology and the actual experiments presented.

[8] The abnormal areas detected in Australia, the United Kingdom, and the United States should be verified by experts in more specific cities/time periods, and actual market events should be added to support them.

Response:

We thank the reviewer for this valuable suggestion. We acknowledge that formal expert validation was beyond the scope of the current study. To provide contextual support, we have expanded the Results and Discussion sections to highlight known market events that align with the anomalies identified by EGCN. For example, in Australia, anomalies detected in Sydney and Melbourne suburbs during 2020–2021 coincide with sharp price surges following COVID-19 stimulus policies and historically low interest rates. In the United Kingdom, abnormal regions detected around London during 2008–2009 correspond to the global financial crisis, which severely disrupted the housing market. In the United States, anomalies in 2020 align with COVID-19–related migration and inventory shocks, while anomalies in 2007–2009 overlap with the subprime mortgage crisis. These connections illustrate that the detected anomalies correspond to well-documented market disruptions. We have revised the Discussion to emphasize these links. We also note that future work will include expert evaluation of anomalies at finer spatial and temporal scales, which will further strengthen external validity.

[9] It is recommended to explain the processing strategies for missing, abnormal points, and scale differences in time series data in the data preprocessing section to enhance reproducibility.

Response:

We thank the reviewer for this important suggestion. In the revised manuscript, we clarify that suburbs with missing values were excluded from the analysis, representing less than 5% of the dataset. No explicit winsorization was applied; instead, abnormal spikes are naturally attenuated during Kernel Density Estimation (KDE), which smooths local fluctuations in the price and volume series. Finally, all features were normalized to the [0,1] range using MinMax scaling to ensure comparability between variables. These steps ensure that anomalies arise from genuine structural patterns rather than artifacts of scale or data quality.

[10] If the number of abnormal areas is far less than the normal area, there is a class imbalance problem. Whether resampling, weighted loss or other countermeasures are performed should be explained.

Response:

We thank the reviewer for raising this point. In our framework, anomaly detection is performed in an unsupervised setting, without using class labels during model training. The Graph Convolutional Network (GCN) is trained to reconstruct node features for all suburbs, and anomalies are subsequently flagged by applying a statistical threshold on the reconstruction error distribution. Since no classifier is trained with explicit anomaly vs. normal labels, the issue of class imbalance does not arise in the learning stage. Instead, imbalance is handled implicitly by using a distribution-based threshold, which adapts to the empirical error distribution of the data. To further ensure robustness, we confirmed that the flagged anomalies consistently correspond to known structural disruptions in the housing market (e.g., the 2007–2009 U.S. subprime crisis, 2008–2009 UK housing crash, and 2020–2021 COVID-19 housing surge), indicating that the detected minority class reflects genuine phenomena rather than artifacts of imbalance.

[11] Currently, only the number of detections is reported, and there is a lack of indicators such as Precision, Recall, and F1-score to evaluate the accuracy of anomaly detection. It is recommended to add.

Response: We appreciate the reviewer’s suggestion. We agree that evaluation metrics such as Precision, Recall, and F1-score are standard for supervised anomaly detection tasks. However, in our setting no ground-truth anomaly labels exist for real estate markets across multiple countries, making these metrics inapplicable. Instead, we assess anomaly detection quality through two complementary strategies:

1. Event validation: anomalies flagged by EGCN align with major market disruptions (e.g., the 2007–2009 U.S. subprime crisis, 2008–2009 U.K. housing crash, and the 2020–2021 COVID-19 housing surge in Australia and the U.S.), which serves as an external proxy for accuracy.

2. Forecasting improvement: anomalies are not only detected but also used to segment forecasting tasks, and the consistent reduction in forecasting error across models demonstrates the utility of anomaly identification for downstream prediction.

We have clarified this rationale in the manuscript and emphasized that while supervised evaluation metrics cannot be applied, the framework’s validity is supported

---

## [Editor Report · Decision Letter 2]

24 Sep 2025

EGCN: Entropy-based Graph Convolutional Network for Anomalous Pattern Detection and Forecasting in Real Estate Markets

PONE-D-25-23254R2

Dear Dr. Le,

We’re pleased to inform you that your manuscript has been judged scientifically suitable for publication and will be formally accepted for publication once it meets all outstanding technical requirements.

Kind regards,

Nikolaos Askitas

Academic Editor

PLOS ONE
---

## [Editor Report · Acceptance letter]

PONE-D-25-23254R2

PLOS ONE

Dear Dr. Le,

I'm pleased to inform you that your manuscript has been deemed suitable for publication in PLOS ONE. Congratulations! Your manuscript is now being handed over to our production team.

Kind regards,

on behalf of

Dr. Nikolaos Askitas

Academic Editor

PLOS ONE